# Structure and function of H$^+$/K$^+$ pump mutants reveal Na$^+$/K$^+$ pump mechanisms

Victoria C. Young [1,6], Hanayo Nakanishi[2,6], Dylan J. Meyer [1],
Tomohiro Nishizawa [3], Atsunori Oshima [2,4,5], Pablo Artigas [1] ✉ &
Kazuhiro Abe [2,4] ✉

Ion-transport mechanisms evolve by changing ion-selectivity, such as switching from Na$^+$ to H$^+$ selectivity in secondary-active transporters or P-type-ATPases. Here we study primary-active transport via P-type ATPases using functional and structural analyses to demonstrate that four simultaneous residue substitutions transform the non-gastric H$^+$/K$^+$ pump, a strict H$^+$-dependent electroneutral P-type ATPase, into a bona fide Na$^+$-dependent electrogenic Na$^+$/K$^+$ pump. Conversion of a H$^+$-dependent primary-active transporter into a Na$^+$-dependent one provides a prototype for similar studies of ion-transport proteins. Moreover, we solve the structures of the wild-type non-gastric H$^+$/K$^+$ pump, a suitable drug target to treat cystic fibrosis, and of its Na$^+$/K$^+$ pump-mimicking mutant in two major conformations, providing insight on how Na$^+$ binding drives a concerted mechanism leading to Na$^+$/K$^+$ pump phosphorylation.

Throughout evolution, ion transport across biological membranes has used classes of proteins with conserved architecture, but varying ion selectivity. The difference in Na$^+$ and H$^+$ selectivity between prokaryote and eukaryote secondary-active transporters[1,2] or between some P-type ATPase members remains unclear. P-type 2C ATPases are obligatory heterodimers formed by a catalytic α subunit and a auxiliary β subunit. They perform different functions through a nearly identical catalytic cycle (Fig. 1a). The Na$^+$/K$^+$ pump (Na$^+$,K$^+$-ATPase, NKA) is expressed in almost all animal cells, where it establishes the Na$^+$ and K$^+$ gradients used for secondary-active transport (to subsequently maintain homeostasis by uptake of nutrients, extrusion of Ca$^{2+}$ or H$^+$, and cell-volume regulation) or for cellular excitability (by the ion channels that dissipate these ion gradients). There are two proton pumps (H$^+$,K$^+$-ATPase, HKA); the gastric proton pump (gHKA) acidifies the gastric-lumen fluid to aid in digestion[3,4], whilst the non-gastric (ng)HKA participates in K$^+$ reabsorption by the colon[5] and the kidney[6], and contributes to the acidification of the airways, a process promoting chronic respiratory infections in the pig model of cystic fibrosis[7]. While

the sodium pump is electrogenic, generating an electric current as it extrudes three Na$^+$ and imports two K$^+$ per ATP hydrolyzed (Fig. 1a, black ions), the gHKA and ngHKA export H$^+$ and import K$^+$ in an electroneutral fashion (Fig. 1a, cyan ions). The exquisite ion selectivity and stoichiometry characteristics of these pumps are determined by their ion-binding sites, formed between transmembrane helixes TM4-TM8 of the catalytic subunit. The four human NKA α subunits (coded by genes *ATP1A1-ATP1A4*) have tissue-specific expression, but identical ion-binding sites (Fig. 1b). In contrast, the ion-binding site residues of the α subunits of gHKA (coded by *ATP4A*) and ngHKA (coded by *ATP12A*) are slightly different (Fig. 1b), suggesting these pumps may have distinct selectivity or stoichiometry.

NKA structures identify three ion-binding sites. Site I and site II prefer to bind Na$^+$ in E1[8], or K$^+$ in E2[9,10], whereas site III exclusively binds Na$^+$ [8]. Instead, E2-structures of the gHKA identify a single K$^+$ occluded at a site overlapping with NKA's site II[11] (throughout this article, the NKA site nomenclature is used). Structures of the gHKA in E1 or of the ngHKA in any state have not been reported.

[1]Department of Cell Physiology and Molecular Biophysics, Center for Membrane Protein Research, Texas Tech University Health Sciences Center, Lubbock, TX, USA. [2]Cellular and Structural Physiology Institute, Nagoya University, Nagoya 464-8601, Japan. [3]Graduate School of Medical Life Science, Yokohama City University, Tsurumi, Yokohama 230-0045, Japan. [4]Graduate School of Pharmaceutical Sciences, Nagoya University, Nagoya 464-8601, Japan. [5]Institute for Glyco-core Research (iGCORE), Nagoya University, Nagoya 464-8601, Japan. [6]These authors contributed equally: Victoria C. Young, Hanayo Nakanishi. ✉e-mail: pablo.artigas@ttuhsc.edu; kabe@cespi.nagoya-u.ac.jp

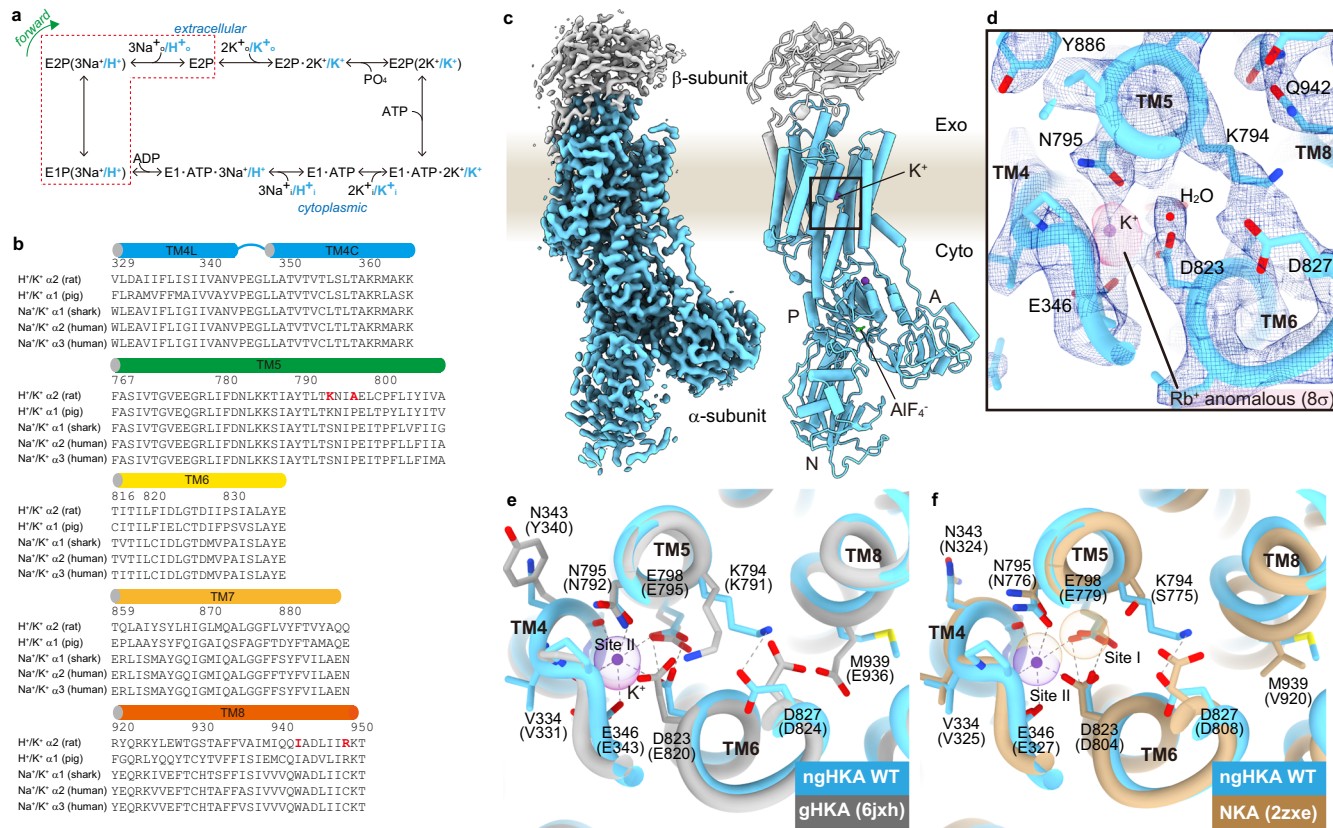

**Fig. 1 | Catalytic cycle and crystal structure of non-gastric H⁺,K⁺-ATPase (ngHKA). a** Post-Albers reaction scheme of H⁺,K⁺-ATPase (HKA, cyan) and Na⁺,K⁺-ATPase (NKA, black). **b** Sequence alignment of the transmembrane ion-binding sites-forming helices in P-type 2C ATPases, TM4-TM8. Amino acids mutated in this study are shown as red. **c** Electron density map (surface, 2σ) and cartoon model (cylinders) of the ngHKA in the AlF₄⁻-inhibited, K⁺-occluded state, mirroring the (K⁺)E2-Pᵢ state of the catalytic cycle. The α- and β-subunits are shown in cyan and gray, respectively. K⁺ ions (one within the membrane domain, one in cytoplasmic domains) are shown as purple spheres. The cytoplasmic A-, P-, and N-domains are also indicated. **d** Close-up of the cation-binding region (enclosed by a box in **c**) viewed from the cytoplasmic side of the membrane. Blue mesh and magenta surface represent 2F₀-F_c electron density map and Rb⁺ anomalous difference Fourier map contoured 2σ and 8σ, respectively. **e** Superposition of the (K⁺)E2-Pᵢ crystal structures of ngHKA (cyan) and gHKA (gray, pdb-code 6jxh, amino acids are indicated in parentheses). **f** Superposition of the (K⁺)E2-Pᵢ ngHKA (cyan) and the (2 K⁺)E2-Pᵢ NKA crystal structure (wheat, pdb-code 2zxe, amino acids are indicated in parentheses). Dotted lines indicate polar interactions within 3.5 Å in the ngHKA in both (**e**, **f**).

Here, we report the crystal structure of wild-type ngHKA in the (K⁺) E2-Pᵢ conformation and show, by functional and cryo-EM structural analyses, that simultaneous substitution of four residues suffices to transform the (1H⁺:1K⁺:1ATP) ngH⁺/K⁺ pump into the (3Na⁺:2 K⁺:1ATP) Na⁺/K⁺ pump.

## Results

### Structural analysis of wild-type non-gastric H⁺,K⁺-ATPase

We co-expressed the rat ngHKA α subunit with the rat NKA β1 subunit in the HEK293S GnTI⁻ cell line (see "Methods"). Purified proteins were crystallized in the presence of the phosphate analog AlF₄⁻, and either K⁺ or its congener Rb⁺. The K⁺-occluded E2-Pᵢ state was solved at 3.3 Å resolution (Fig. 1c, d and Supplementary Table 1). The structure of this important drug target reveals mechanistic differences between this pump and the gastric proton pump. The position of the intracellular domains and transmembrane segments (Supplementary Fig. 1) were almost identical (RMSD = 1.79 Å) to those previously reported on the Tyr799Trp-gHKA arrested in the same conformation (Fig. 1e, PDB code 6jxh, the Tyr799Trp was used to stabilize the occluded conformation in ref. 11). A strong density is observed at ion-binding site II and a weaker density, presumably a water molecule, close to Asp823 (zoomed-in view, Fig. 1d). The anomalous difference Fourier map calculated with the Rb⁺-bound crystal shows a unique strong peak overlapping the density at site II, unambiguously identifying this density as a bound K⁺. The occluded K⁺ is coordinated by surrounding oxygen

atoms contributed by side-chain oxygens (Glu346, Asn795, Glu798, and Asp823) and main-chain carboxyl groups (Val341, Ala342, and Val344), giving a near-ideal valence (0.99, Supplementary Table 2). As this pump is electroneutral[12] (see below) this result demonstrates that the ngHKA exchanges 1H⁺ and 1K⁺ per cycle, like the gastric pump[11].

Despite sharing ~70% sequence identity, the α subunits of gHKA and ngHKA show remarkable differences at their ion-binding sites. These variations may relate to both the less stringent selectivity of ngHKA (reportedly able to export Na⁺ or H⁺[13], or to import Na⁺ in lieu of K⁺)[14] and to the much shallower pH gradient across most ngHKA expressing epithelia[7,15] compared to the gastric mucosa[16]. E2-Pᵢ is the conformation reached immediately after H⁺ dissociation in the forward catalytic cycle (Fig. 1a), and these structures provide insight into the different H⁺ extrusion mechanisms by these two proton pumps (Fig. 2). Both pumps have a lysine residue (Lys794 in ngHKA; Lys791 in gHKA, Fig. 1e) at a position where NKA has a serine (Ser775, Fig. 1b, f, NKA numbering refer to the pig α1). This lysine forms a salt bridge with carboxylate residues in TM6: Asp827 in ngHKA versus Glu820 in gHKA. The salt bridge Lys791-Glu820 is essential for gHKA's ability to release H⁺ into the highly acidic stomach (pH -1), a feat achieved by the reduction in Glu820's p$K_a$ due to the 2.5 Å proximity of Glu795, a conserved TM5 residue[17] (Fig. 2, left). The shorter side chain of Asp823 on ngHKA (corresponding to gHKA's Glu820), simultaneously reduces p$K_a$ modulation by Glu798 and the likelihood of ionic interaction with Lys794. In the gHKA, Asp824 (Asp827 in ngHKA) hydrogen bonds to

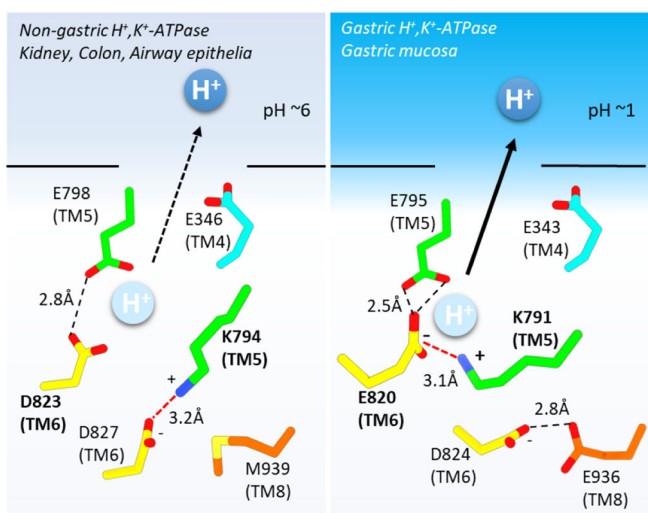

**Fig. 2 | Schematic model for proton extrusion for ngHKA and gHKA.** See text for details.

TM8's Glu936, facing away from the cation-binding site, thus preventing its interaction with Lys791 (Fig. 2, right). In contrast, the ngHKA has Met939, and the lack of a hydrogen-bond partner makes Asp827 face towards the ion-binding site, forming a salt bridge with Lys794 which now faces to the opposite side of TM6 (Figs. 1e and 2). Thus, the distinct cation-binding-site structures suggest that the ngHKA has evolved to produce smaller reductions in extracellular pH, as suggested by the higher pH observed in epithelia expressing this pump. This may reflect that the main physiological role of the ngHKA is to reabsorb K[+5] instead of building H[+] gradients. However, the proton gradient built by the ngHKA is a major contributor to the development of cystic fibrosis infections, due to the impaired bicarbonate transport by CFTR[7,18], which neutralizes ngHKA-induced acidification in non-CF patients. The ngHKA structure provides a template for future development of desired ngHKA-specific drugs.

## Lys794 mutants of the non-gastric H⁺,K⁺-ATPase

Mutagenesis pinpointed Lys794 as the critical residue for ngHKA's electroneutrality because its mutant Lys794Ala, and to a lesser extent Lys794Glu, are electrogenic[19]. We evaluated the enzymatic, electrophysiological, and structural characteristics of wild-type (WT), Lys794Ser and Lys794Ala ngHKA (Fig. 3). ATPase activity (Fig. 3a and Supplementary Fig. 2a) was measured as described[20] (see "Methods") in crude membrane preparations from HEK293S cells expressing wild-type (WT, blue), Lys794Ser (K794S, dark yellow), and Lys794Ala (K794A, green) ngHKA. All three constructs showed K⁺-dependent ATPase activity in the absence of Na⁺ (Fig. 3a, open symbols), demonstrating H⁺,K⁺-ATPase activity. Adding Na⁺ to the reaction media caused mutant-specific effects (Fig. 3a and Supplementary Fig. 2a). First, Na⁺ augmented the activity of both mutants (Lys794Ser at [K⁺] ≥ 0.5 mM, and Lys794Ala at [K⁺] ≥ 10 mM), but not of WT pumps. Lack of Na⁺ activation demonstrates that Na⁺ is a poor surrogate of H⁺ for WT-ngHKA, as previously reported[14,21]. The Na⁺-augmented activity of both mutants suggests that their phosphorylation step is faster when Na⁺ interacts with the intracellular-facing site(s) than when H⁺ is the only interacting cation. These mutations also altered ion interactions at the extracellular-facing sites where K⁺ binding accelerates dephosphorylation, as the presence of Na⁺ increased the $K_{0.5,K⁺}$ of Lys794Ala, and to a lesser extent of WT, but did not alter it in Lys794Ser. Therefore, both mutations eliminated H⁺-exclusive dependence and altered Na⁺/K⁺ competition, but both failed to make a strict Na⁺/K⁺ pump.

Electrogenic reactions were studied in *Xenopus* oocytes with a two-electrode voltage clamp, as described[22] (see Methods) (Fig. 3b–d). Current traces at −50 mV illustrate the lack of response to the application of 10 mM K⁺ in oocytes expressing WT- and Lys794Ser (due to lack of electrogenic transport), and the dose-dependent activation of outward currents by K⁺ in oocytes expressing Lys794Ala or NKA (Fig. 3b). Ouabain (an NKA inhibitor that also inhibits ngHKA[23]) blocked K⁺-induced responses in both constructs and also caused an inward current deflection in the absence of K⁺ in Lys794Ala, indicating K⁺-independent electrogenic transport. To study the partial reactions in the absence of K⁺, we measured the ouabain-sensitive transient currents (current without ouabain – current with ouabain) elicited by voltage pulses (Fig. 3c). In the NKA, these currents are due to negative voltages forcing Na⁺ ions back into their binding sites favoring E1P(3Na⁺) occupancy (backward direction in Fig. 1a), and positive voltages favoring Na⁺ release to the external side and E2P occupancy. Transient currents were small in WT- and Lys794Ala-injected oocytes, but much larger in NKA- or Lys794Ser-injected ones (Fig. 3c). The charge (Q, from the current integral upon returning to −50 mV) was plotted against the applied voltage (V) (Fig. 3d). The Q–V curves are sigmoidal for NKA (squares) or Lys794Ser (triangles), while the small charge moved by Lys794Ala (diamonds) and WT-ngHKA (circles) lacks sigmoidicity. Thus, Lys794Ala electrogenic transport differs from NKA.

The Cryo-EM structures of Lys794Ala and Lys794Ser in the K⁺-bound E2-Pᵢ were solved at 2.80 and 2.99 Å resolution, respectively (Fig. 3e, f, Supplementary Fig. 3, and Supplementary Table 3). The global structures of these mutants were indistinguishable from WT − ngHKA (Supplementary Fig. 1). Lys794Ala displays one strong spherical density at site II (valence 0.86, Supplementary Table 2), representing K⁺, and few weaker densities around site I, where we modeled waters (Fig. 3e). In contrast, Lys794Ser displays two strong densities at sites I and II (valence 1.03 and 1.07, respectively, Fig. 3f and Supplementary Table 2), representing two bound K⁺ ions, as reported in a similar gHKA mutant[24]. The enzymatic, electrophysiological, and structural characteristics indicate that both mutants change the stoichiometry and selectivity of HKA. While the alanine mutant mediates electrogenic transport of 2Na⁺ or 2H⁺ for 1K⁺, the NKA-mimicking serine mutant mediates electroneutral transport of 2Na⁺ or 2H⁺ for 2K⁺.

## How to make a Na⁺/K⁺ pump

We used electrophysiology to evaluate the characteristics of various combinations of NKA-mimicking mutants in ngHKA (Fig. 4). Oocytes expressing the Lys794Ser/Arg949Cys mutant lacked K⁺-induced (Fig. 4a, b, orange circles) or ouabain-sensitive transient charge movement (Fig. 4c, d), indicating that this mutant is <u>not</u> a prototypical NKA. The ngHKA has Ala797, at a position where both NKA and gHKA have Pro778 and Pro794, respectively. An alanine instead of a proline between three ion-coordinating side chains in TM5 (NKA's Ser775, Asn776, and Glu779) could alter ion interaction. The steady-state (Fig. 4a, b, magenta pentagons) and transient currents (Fig. 4c, d) of the Ala797Pro introduced into the Lys794Ser/Arg949Cys template (SPC mutant) differed from NKA's.

Gln923 and Asp926 contribute Na⁺-coordinating side chains to site III in the NKA[8,25–27] and are conserved in HKA pumps (Gln942 and Asp945, respectively, in ngHKA). Between these two residues, facing the other side of the TM8 helix, NKA has tryptophan (Trp924) while both HKAs have an isoleucine (Ile943 in ngHKA). Mutation to arginine of the equivalent Trp931 in human NKA α1 was found in a patient with a lethal form of hypomagnesemia with seizures[28]. Compared to isoleucine, the sterically larger and hydrogen-bond-capable tryptophan side chain may help stabilize the site-III-coordinating residues in TM8. Therefore, we introduced Ile943Trp in the Lys794Ser/Arg949Cys template (SWC mutant). SWC showed outward currents when exposed to K⁺, but the subsequent application of ouabain without K⁺ also caused an outward deflection of the current (Fig. 4a). The ouabain-

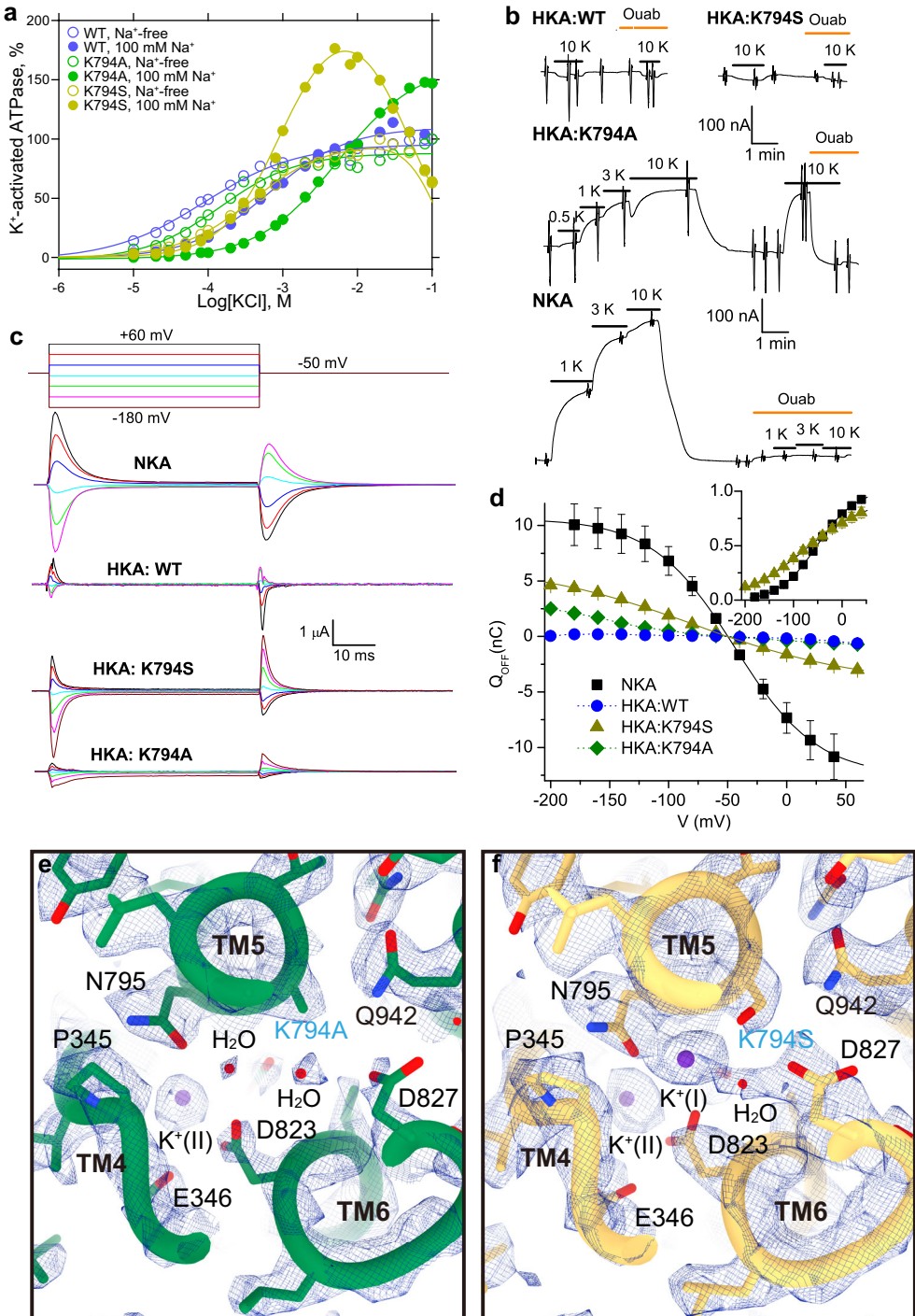

**Fig. 3 | Functional and structural characterization of WT-, K794S-, and K794A-ngHKA. a** Representative results from more than three independent measurements of K⁺-dependent ATPase in crude membrane fractions from cells expressing ngHKA WT (blue), K794A (green) and K794S (yellow), measured in the absence (open symbols) or in the presence (filled symbols) of 100 mM Na⁺. The maximum K⁺-activated activity without Na⁺ was set as 100% and that without K⁺ or Na⁺, as blank. Line plots are Hill fits (see "Methods"). **b** Current at −50 mV from Na⁺-loaded oocytes expressing WT-, K794A-, and K794S-ngHKA or NKA pumps. Application of K⁺ activated outward currents only in oocytes expressing K794A-ngHKA ($K_{0.5} = 2.8 \pm 0.7$ mM, $nH = 1.75$, $n = 5$) or NKA ($K_{0.5} = 1.21 \pm 0.34$ mM, $nH = 1.47$, $n = 9$). Note similar $K_{0.5}$ for current and ATPase activation at 100 mM Na⁺ for Lys794Ala. Ouabain (25 mM ngHKA & 10 mM NKA) blocks subsequent responses to K⁺. **c** Ouabain-sensitive transient currents elicited by the pulse protocol shown on

top. The protocol was repeated before and after the application of ouabain (10 mM for NKA, 25 mM for ngHKA mutants) to obtain the pump-specific signals displayed (current without ouabain minus current in ouabain). **d** Mean Q–V from experiments like those in (**c**). The Boltzmann fitted to individual experiments had $V_{1/2} = -75 \pm 2$ mV and $kT/z_q e = 64 \pm 3$ mV ($n = 16$) for Lys794Ser (yellow line), and $V_{1/2} = -51 \pm 1$ mV and $kT/z_q e = 40 \pm 1$ ($n = 12$) for NKA. Data for Lys794Ala ($n = 7$) and WT ($n = 4$) could not be fitted. **e** Ion-binding sites of K794A-ngHKA with a K⁺ ion modeled at site II. Water was modeled in the weak density near site I (side-chain oxygens of Asn795 (2.30 Å) and Asp823 (2.39 Å) are too close to accommodate K⁺). **f** Ion-binding sites of K794S-ngHKA. Strong densities are consistent with K⁺ ions bound at both sites. Both K⁺-occluded structures are viewed from the cytoplasmic side.

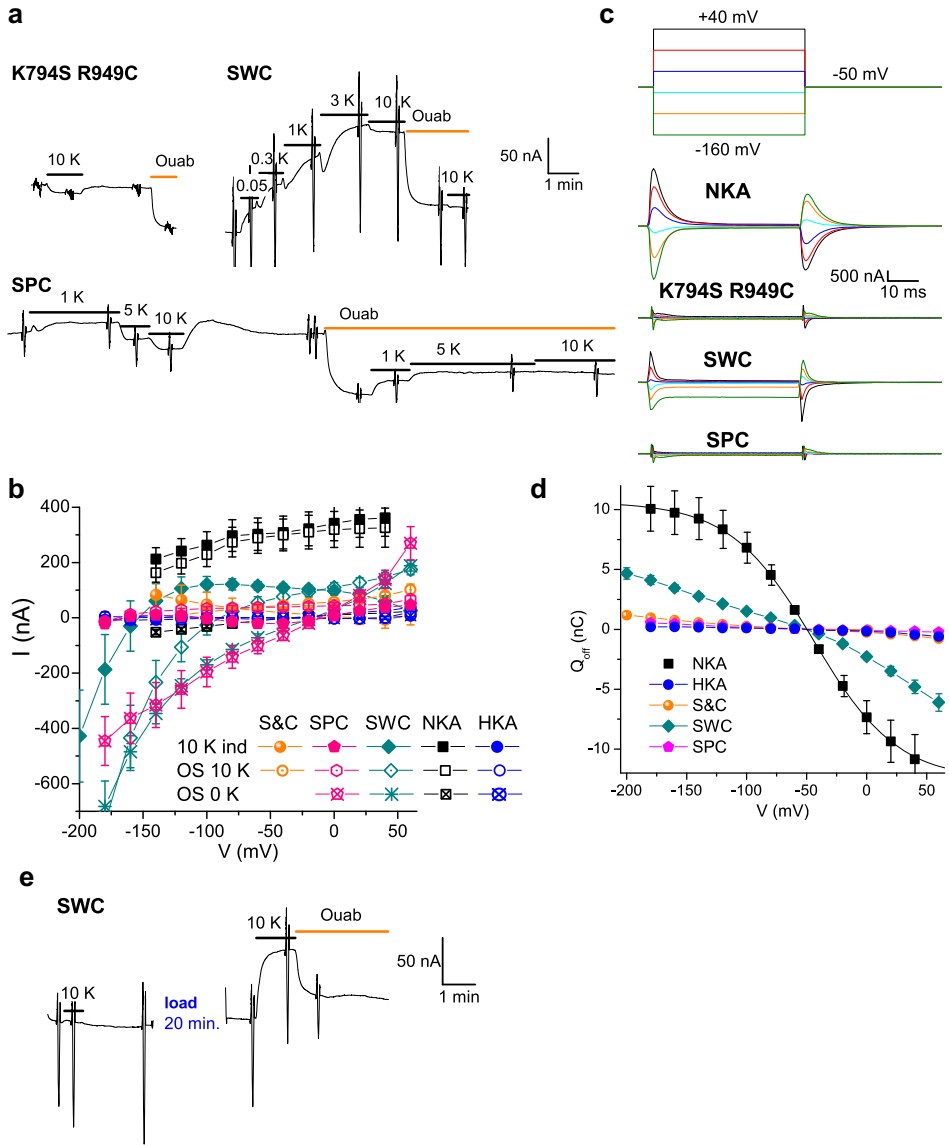

**Fig. 4 | Functional analysis of "intermediate" ngHKA mutants. a** Current at −50 mV from a Na⁺-loaded oocytes expressing K794S/R949C-, SWC-, or SPC-ngHKA. Application of K⁺ induced a small inward current in oocytes expressing K794S/R949C-ngHKA, whereas it activated outward currents in SWC-ngHKA expressing oocytes. In contrast, SPC expressing oocytes showed small outward currents at low (≤1 mM) K⁺, but large dose-dependent inward currents at higher [K⁺]. Ouabain application causes an inward deflection of the current in K794S/R949C-, and SPC-ngHKA, indicating that these mutants perform an electrogenic transport mode without K⁺. Vertical deflections correspond to the application of 50-ms long pulses (seen at a slow sampling rate) to voltages ranging between −160 and +40 mV to construct I–V plots. **b** Mean steady-state I–V for the 10 mM K⁺-induced current (solid symbols), the ouabain-sensitive current in 10 mM K⁺ (open symbols) and the ouabain-sensitive current without K⁺ (crossed symbols) for K794S/R949C- (orange circles), SWC- (teal diamonds), SPC-ngHKA (pink hexagons), NKA (black squares), and ngHKA (blue circles). **c** Ouabain-sensitive transient currents in response to the pulse protocol shown on the top, in oocytes bathed in Na⁺ solution without K⁺. The current measured when the pulse returned to −50 mV was integrated to construct the charge–voltage curves. **d** Mean Q–V curve for K794S/R949C (orange circles, $n = 10$), SPC-ngHKA (pink hexagons, $n = 5$), and SWC (teal diamonds, $n = 8$). These Q–Vs could not be fitted with Boltzmann distributions. **e** Effect of intracellular Na⁺ depletion, followed by Na⁺ loading in an oocyte injected with SWC-ngHKA. The oocyte was incubated for 1 h in Na⁺-depletion, K⁺-containing, solution before recording ("Methods"). Following the first application of 10 mM K⁺ (which failed to activate current) the clamp was turned off and the oocyte perfused with Na⁺-loading solution for 20 min. The clamp was turned on again and K⁺ applied once more, activating outward current. Thus, intracellular Na⁺ is required for electrogenic transport by SWC.

sensitive current in K⁺ is outward at voltages >−80 mV (Fig. 4b, open diamonds), but smaller than the K⁺-induced current (Fig. 4b, solid diamonds), indicating that in addition to electrogenic cycling, K⁺ inhibits a large inward current (Fig. 4b, stars). This ouabain-sensitive inward current is increased at very negative voltages (Fig. 4b, c) and resembles currents observed in certain NKA mutants that reduce apparent affinity for Na⁺ (refs. 27, 29, 30). SWC's non-sigmoidal Q–V curve (Fig. 4d, diamonds) indicate that this mutant may have an extreme reduction in the affinity for external Na⁺. To confirm that SWC

electrogenic function requires the presence of intracellular Na⁺, we studied the effect of Na⁺ depletion on the K⁺-activated currents (Fig. 4e). An SWC-injected oocyte depleted of intracellular Na⁺ by a one-hour incubation in a Na⁺-free, Na⁺-depletion solution (see "Methods"), lacked K⁺-induced outward currents. Subsequently, the clamp was turned off for 20 min (axis break) while the oocyte was incubated in Na⁺-loading solution. After the voltage clamp was turned on again, application of K⁺-induced outward currents, demonstrating that this mutant requires intracellular Na⁺ to function.

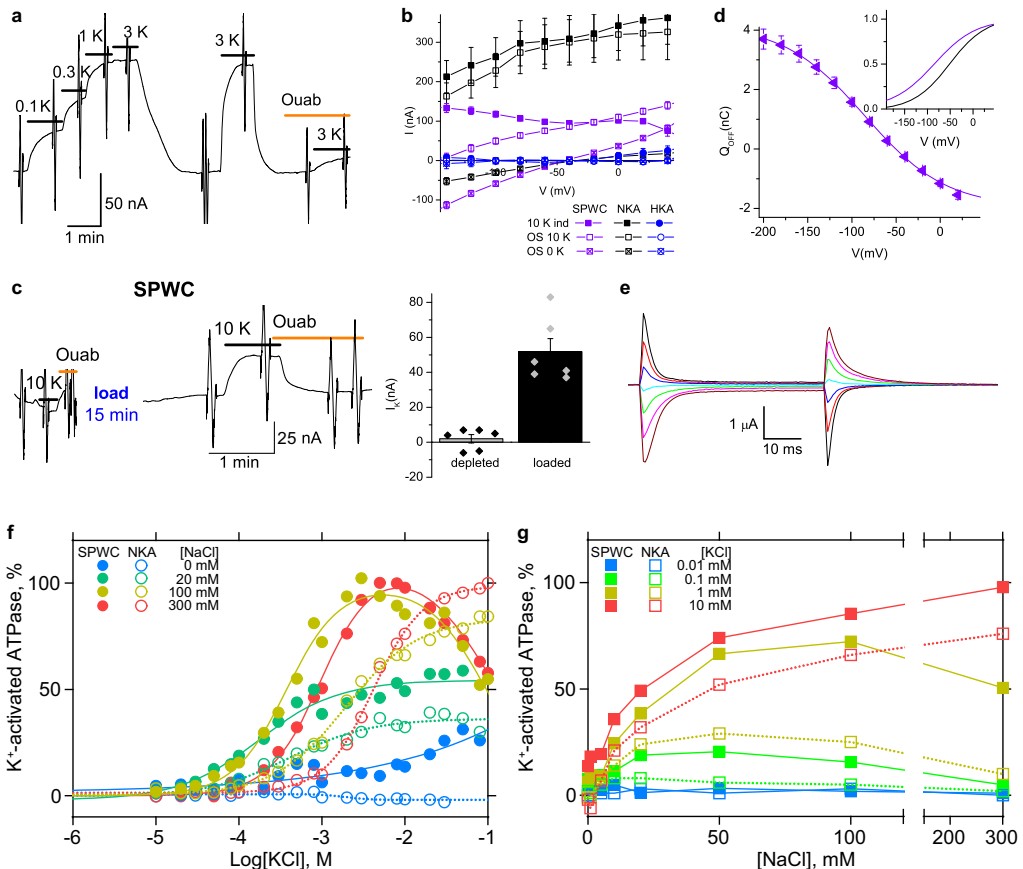

**Fig. 5 | Functional and structural characterization of NKA-like SPWC-ngHKA.**
**a** Dose-dependent activation of outward current by K$^+$ in a Na$^+$-loaded oocyte expressing SPWC-ngHKA. Application of 25 mM ouabain blocked the effect of subsequent K$^+$ applications. **b** I–V for K$^+$-induced current (solid purple squares) and for the ouabain-sensitive (OS) current measured with 10 mM K$^+$ (open purple squares) or without K$^+$ (crossed squares). Black squares and blue circles are results for NKA and WT-HKA, respectively, in the same conditions. **c** Left, Application of 10 mM K$^+$ on a Na$^+$-depleted oocyte expressing SPWC-ngHKA did not activate outward current at −50 mV. In contrast, K$^+$ application following a 15-min long exposure to Na$^+$-loading solution (during the break in the trace) activated outward current. Right, average change in K$^+$-induced current in Na$^+$-depleted and Na$^+$-loaded oocytes. **d** Ouabain-sensitive transient currents measured as in Fig. 2c, from an oocyte expressing SPWC-ngHKA. **e** Mean Q–V curve from 14 experiments,

the purple line is the fit to a Boltzmann distribution to the average data (from −200 to 0 mV to avoid a linear component at positive voltages) with $V_{1/2}$ = −85 mV, $kT/z_qe$ = 46 mV. Inset, Q–V normalized to total charge from Boltzmann fits, for SPWC and NKA. **f** [K$^+$]-dependent ATPase activity in membrane preparations expressing SPWC-ngHKA (solid) or NKA (open) at 0 (blue), 20 (green), 100 (yellow), and 300 (orange) mM Na$^+$. The line plots are fits of a double Hill equation to the data (see Supplementary Fig. 4 and Methods). Note that SPWC's activity without Na$^+$ is very small and that it does not saturate, even at 100 mM K$^+$ (despite the absence of the competitor Na$^+$). Because at these high concentrations, K$^+$ inhibits ATPase in the presence of Na$^+$, this Na$^+$-independent activity must reflect non-specific ATPase activity. **g** [Na$^+$]-dependent ATPase activity for SPWC-ngHKA (solid) or NKA (open) at 0.01 (blue), 0.1 (green), 1 (yellow) and 10 mM K$^+$ (orange). Representative results from more than three independent measurements are shown in (**f**, **g**).

Addition of Ala797Pro on the SWC template (SPWC mutant) generates a archetypal Na$^+$/K$^+$ pump (Fig. 5). The K$^+$-concentration-dependent activation of outward current at −50 mV illustrates that SPWC has higher apparent affinity for K$^+$ (Fig. 5a, $K_{0.5,K+}$ = 0.21 ± 0.03 mM, $nH$ = 1.3, $n$ = 13) than NKA (1.21 ± 0.34 mM, $nH$ = 1.47, $n$ = 9). The I–V curves (Fig. 5b) illustrates that K$^+$-induced currents (solid squares) are larger than ouabain-sensitive currents in saturating K$^+$ (open squares) at negative voltages, but smaller at positive voltages. This is because SPWC displays an ouabain-sensitive "leak" in both inward and outward directions (crossed squares). Therefore, the K$^+$ induced current at negative voltages is the sum of the normal pump current activated, and the leak current inhibited by K$^+$. However, at positive voltages, the smaller increase in outward current reflects that the pump current acts on top of an outward "leak" that disappears as the pump engages in Na$^+$/K$^+$ transport. The ouabain-sensitive current in K$^+$ (open squares) probably represents the real pump current, but we cannot distinguish the two types of outward currents at positive voltages. The K$^+$-induced and ouabain-sensitive currents in K$^+$ match at −50 mV. To confirm that SPWC mimics other characteristics of NKA, we studied whether K$^+$-induced currents require the presence of intracellular Na$^+$ (Fig. 5c) as before for

the SWC mutant (Fig. 4e). The current traces illustrate that application of K$^+$ failed to activate current after Na$^+$ depletion, but activated outward current after Na$^+$ loading, in both oocytes expressing the SPWC mutant or NKA (Fig. 5c, with results summarized in the bar graph on the right). We also evaluated the ouabain-sensitive transient currents of SPWC (Fig. 5d), which are very similar to those in NKA (Fig. 3c). The slower kinetics of current induced by negative-going pulses in SPWC-oocytes compared to NKA-oocytes, indicate a reduced apparent affinity for extracellular Na$^+$, which is reflected in a leftward shift of the Q–V curve for the mutant (Fig. 5e, triangles, $V_{1/2}$ = −85.5 ± 7.3 mV, $n$ = 14) with respect to that for NKA (Fig. 5e, black line, $V_{1/2}$ = −51 ± 1 mV ($n$ = 12)), indicating a 2.7-fold reduced apparent affinity for extracellular Na$^+$ compared to NKA (25 mV/twofold reduction[31,32]). This reduced apparent affinity for extracellular Na$^+$, which competes with K$^+$ for extracellular sites, may contribute to the larger $K_{0.5,K+}$ of this mutant.

The Na$^+$- and K$^+$-concentration dependencies of ATPase activity were measured in crude membrane preparations from HEK293S cells expressing SPWC-ngHKA (solid symbols) or NKA (open symbols) (Fig. 5f, g and Supplementary Fig. 4). The K$^+$-dependence for SPWC is biphasic, whereas that for NKA is not. The activation phase with 100 mM

Na$^+$ (yellow circle, $K_{0.5,K^+} = 0.38$ mM) has identical half-maximal activation to the activation of outward currents in the presence of comparable Na$^+$ concentrations (125 mM). The inhibitory phase of Na$^+$,K$^+$-ATPase activity at high K$^+$ concentrations (Fig. 5f, IC$_{50,K^+} = 96$ mM) resembles those reported in mutants that disrupt Na$^+$ affinity[26,27,33,34], and probably reflects a combination of increased K$^+$ affinity and reduced affinity for the competing Na$^+$ for the pump in the E1 conformation. In contrast to the high Na$^+$-independent activities observed in other ngHKA constructs (Fig. 3a), NKA and SPWC preparations lack Na$^+$-independent, K$^+$-dependent, ATPase activity (Fig. 5f, open circles). The Na$^+$-dependence of ATPase activity and electrogenic properties of SPWC match those of a genuine NKA (Fig. 5g). Therefore, all functional results indicate SPWC is an obligatory Na$^+$- and K$^+$-dependent electrogenic ATPase.

To define the cation-binding stoichiometry of the SPWC mutant, we determined its cryo-EM structures in the presence of Na$^+$ and AMPPCP, to obtain the Na$^+$- and AMPPCP-bound E1 state, at 3.08 Å resolution (Fig. 6a, Supplementary Fig. 3, and Supplementary Table 3, see Fig. 6e for relative orientation of SPWC mutation and cation-binding site) and in the presence of K$^+$ and AlF$_4$ to obtain the K$^+$-bound E2 state, at 3.26 Å resolution (Fig. 6b and Supplementary Fig. 3). The overall molecular conformation of the 3Na$^+$·E1-ATP is strikingly close to that of the NKA in the (3Na$^+$)E1P-ADP state[8] (pdb-code 3wgu, Supplementary Fig. 5). And the (2 K$^+$)E2-AlF state is nearly identical to the (2 K$^+$)E2-MgF state of NKA[9,10] (pdb-code 2zxe, Supplementary Fig. 5). A close-up view shows that the SPWC mutant binds three Na$^+$ ions in E1 (Fig. 6c), and two K$^+$ ions in E2 (Fig. 6d), like the NKA. Small spherical densities were identified in the cation-binding region of the 3Na$^+$·E1-ATP state. Due to the weak densities at sites I and II, the Na$^+$ at site III is the only one unambiguously determined, while it is difficult to discriminate the other two Na$^+$ from bound water molecules. Utilizing the previously reported NKA crystal structure[8] (PDB 3wgu) as a reference, we modeled three Na$^+$ ions in our EM density map, and water molecule for other spherical densities (orange and red spheres, respectively, Fig. 6c). The densities were interpreted by considering optimal water-cation distances, hydrogen bonds, electrostatic repulsion, and the need for multiple oxygens surrounding a cation. This results in a close overlap between the Na$^+$ modeled in our structure with those in the NKA crystal structure (offset, respectively, by 0.5, 1.6, and 0.3 Å for sites I, II, and III, Fig. 7). Thus, we determined two cornerstone intermediates of the transport cycle of the SPWC-ngHKA, demonstrating that it has 3Na$^+$:2K$^+$ stoichiometry, like the NKA.

## Discussion

Comparison between the SPWC structures in the 3Na$^+$·E1-ATP and (2K$^+$)E2-P$_i$ states reveals some key conformational changes accompanying the ATP hydrolysis-coupled cation transport by the SPWC mutant. After K$^+$ dissociation in the E1 state, three Na$^+$ ions enter the TM domain to bind sequentially to site III, site I and site II, as elegantly proposed in the atomic model of cooperative binding of ref. [8], where binding to each site induces the formation of the next site. The last Na$^+$-binding step at site II in E1-ATP requires coordination by Glu346 (NKA's Glu327), an essential residue for the gating mechanism that occludes ions from the cytoplasmic side (Fig. 8). When comparing this conformation to the E2 state structure, it is clear that Na$^+$ coordination at site II needs to be accompanied by a large conformational change of the SPWC structure (Fig. 8 and Supplementary Movie 1), resembling the changes occurring in NKA (Supplementary Fig. 5). Positioning of Glu346 requires the large vertical and tilting movement of TM4 (Fig. 8a, c) which pulls TM3, TM2, and TM1 with it (as reported in SERCA[35] and NKA[8,36]). TM1-TM3 movement drives the large downward and rotating displacement of the A-domain (Supplementary Movie 1 and Fig. 8b, d). More importantly, the P-domain connected to TM4 and TM5 is displaced by the vertical and tilting displacement of TM4 and by the straightening of TM5 (Fig. 8e and Supplementary Movie 1). Such straightening, also seen in E1 conformations of SERCA (pivoting at TM5

Gly770)[35] and the lipid flippase ATP11C[37] (pivoting at TM5 Lys880), is essential to bring the P-type ATPase conserved phosphorylation site aspartate, Asp388, in close proximity to the γ-phosphate of the ATP bound to the N-domain, triggering autophosphorylation (Supplementary Movie 1). We found that TM5 pivots at site III (Fig. 8e), where the Na$^+$ located on the extracellular side of Tyr790 is coordinated by side-chain oxygens from Thr793 (2.3 Å), Ser794 (3.7 Å), Asp827 (3.3 Å), Gln942 (2.5 Å), and Asp945 (3.5 Å) and by the main-chain carboxyl groups Tyr790 (2.8 Å) and Thr791 (4.2 Å) (Supplementary Table 2), nearly identical to its coordination in NKA[8] (Fig. 7). Coordination by the main-chain carbonyls of Tyr790 and Thr791 breaks the alpha-helical hydrogen-bond network of TM5. This causes the distance between the main-chain carbonyl of Tyr790 and the side chain of the serine introduced at 794 to be slightly longer in the 3Na$^+$·E1-ATP (5.5 Å) than in the E2-P$_i$ state (4.3 Å). This difference results in the subtle 1.1 Å shift in the main-chain trace around site III causing a 5.6 Å displacement in the cytoplasmic end, moving the P-domain as required for autophosphorylation (Fig. 8e). Therefore, we propose that in addition to providing a mechanism for Na$^+$ selectivity at site III by the size restriction as previously proposed[8], the straightening of TM5 is essential for the allosteric conformational change of the whole enzyme structure required for phosphorylation. Note that straightening alone does not allow autophosphorylation, as the A-domain probably blocks access of the N-domain to the P-domain until binding to sites I and II occurs.

Both SWC and SPWC mutants are electrogenic, confirming that introduction of tryptophan at position 943 was essential to generate site III. This result (unexpected given that the side chain of 943 faces the other side of ion-coordinating residues in the TM8 helix) is explained by a hydrogen bond of Ile943Trp with Gln974 in TM9 which participates in a hydrogen-bond network with Tyr790, bridged by a water molecule (Fig. 8f). A similar hydrogen-bond network is present between the corresponding residues of NKA (Glu954 and Tyr771, Fig. 7a). Before determination of the Na$^+$-bound crystal structures, it was proposed that the sodium-exclusive site III was formed by the side chains of Glu954 and Tyr771 together with other main-chain carboxyl groups, based on alterations in Na$^+$ affinity observed in several NKA mutations, including Glu954Ala[38,39]. Our findings clarify Na$^+$ coordination revealing the indirect effect of Glu954Ala in NKA, as destruction of the water-mediated hydrogen-bond network with Trp924 eliminates the stabilizing effect of the tryptophan on Gln923 (which also hydrogen bonds to Tyr771[32]) and Asp926, drastically impairing Na$^+$ binding at the real site III. Thus, formation of Na$^+$-exclusive site III in the ngHKA background requires the allosteric influence from the side chain at 943, which is not directly involved in Na$^+$-coordination.

In the (K$^+$)E2-P$_i$ state of ngHKA, Asp945 forms a salt bridge with Arg949 (Fig. 9, an interaction also observed in gHKA's equivalent residues[11]). Thus, Asp945's side chain is neutralized and unavailable to interact with Na$^+$ at site III. Two observations support this interpretation: first, Arg949Cys introduced in the ngHKA's SPW background restores the negative charge of Asp945, allowing it to interact with Na$^+$ (Figs. 6 and 8), and second, the mutation Cys930Arg in NKA eliminates Na$^+$ binding at site III making Na$^+$/K$^+$ pump transport electroneutral[33]. Considering the distances, steric hindrance, and almost identical conformation of Arg949Cys in SPWC to Cys930 in NKA during the E1–E2 transition (Fig. 9, bottom), it seems unlikely that Arg949 eliminates Na$^+$ binding due to direct occupancy of site III by its guanidinium moiety. It is more probable that this reflects charge neutralization of Asp945 as it forms a salt bridge with R949 (a feature observed in both HKA (K$^+$)E2-P$_i$ structures). The neutralizing mutation of the equivalent residue in the human NKA α3 subunit (Asp923Asn) causes rapid onset dystonia parkinsonism and decreases Na$^+$ affinity by >100-fold without affecting K$^+$ affinity, indicating the effect is site-III specific[26]. Free-energy-perturbation-molecular-dynamics simulations also suggest that NKA's Asp926 preference to bind Na$^+$ is reduced by protonation[40,41]. Therefore, it seems likely that Asp945 neutralization

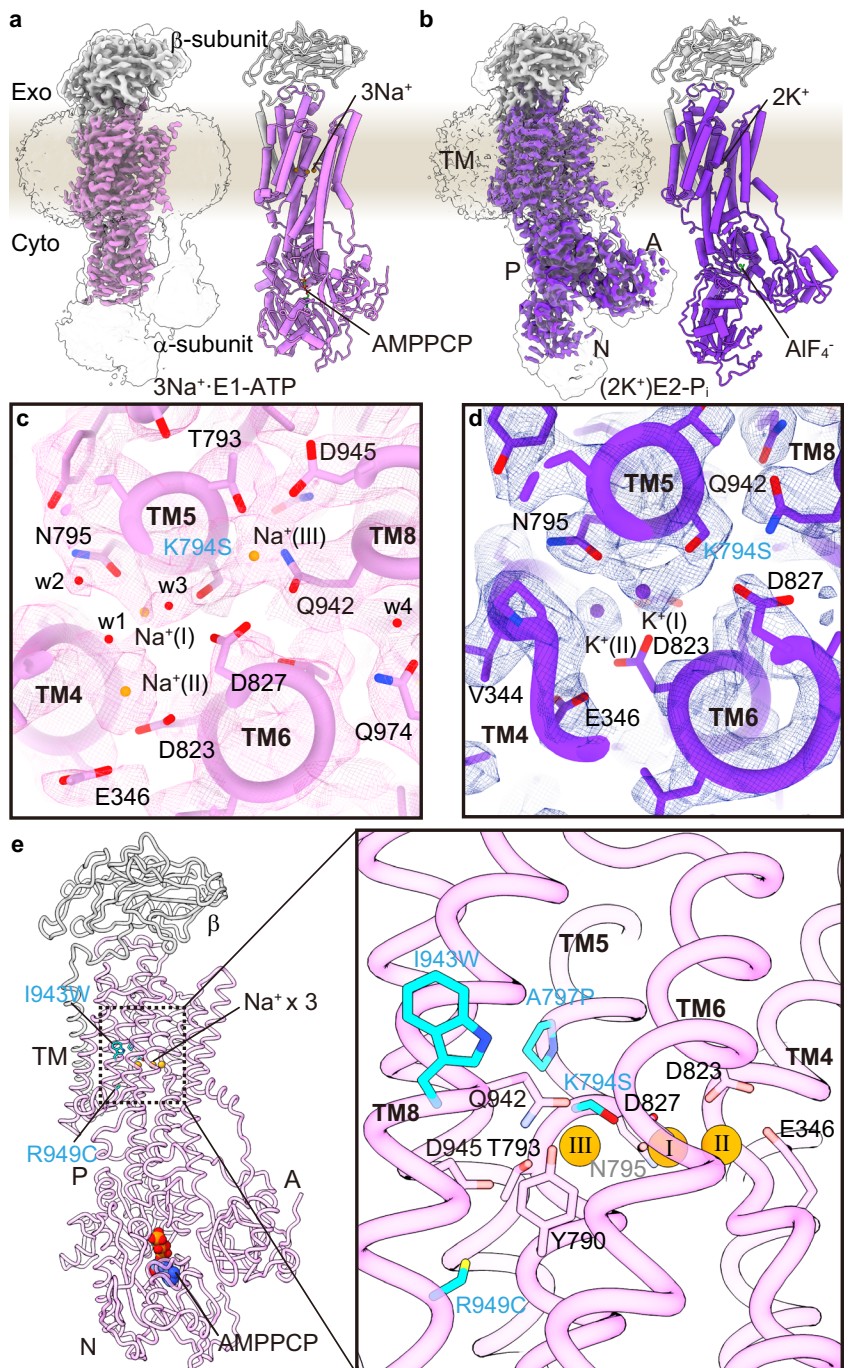

**Fig. 6 | Cryo-EM structure of NKA-like SPWC-ngHKA. a**, **b** EM potential maps (colored surfaces indicate higher contour level, while transparent surfaces indicate lower levels with micelle and cytoplasmic domains visible) and cartoon models of the SPWC-ngHKA mutant in the presence of 300 mM $Na^+$ and 5 mM AMPPCP to induce the $3Na^+ \cdot E1$-ATP state (**a**), and with 100 mM KCl and 1 mM $AlF_4^-$ to lock the pump in the $E2$-$P_i$ state (**b**). The β-subunit is shown in gray and the α-subunit in pink, for E1, and purple, for E2. Bound ions are shown as orange ($Na^+$) or purple ($K^+$) spheres. See Fig. 7 for comparison with NKA structures. **c** Cytoplasmic view of the

ion-binding sites region in E1-ATP showing the densities likely representing the three bound $Na^+$ (orange spheres) and several waters (red spheres, w1-4). **d** Cytoplasmic view of the cation-binding sites in the $(2K^+)E2$-$P_i$ state with two occluded $K^+$ ions (purple spheres). **e** Overall structure of SPWC mutant in $3Na^+ \cdot E1$-ATP state in ribbon representations, and close-up view around its cation-binding site. The four point mutations (cyan) and other amino acids involved in $Na^+$ coordination are displayed as sticks. Bound AMPPCP and the three $Na^+$ ions (site I, II, and III) are shown as spheres.

by interaction with Arg949 in the WT-ngHKA, together with the lack of a stabilizing $H^+$-bond network by the presence of Ile943, prevents $Na^+$ binding at site III.

If protonation/deprotonation is key to the function of site III, the question arises of how Asp945 protonation may be regulated. It is thought that the $H^+$ that impedes $Na^+$ binding in E2P comes from the cytoplasm and then returns to the cytoplasm when the pump is in the

E1 conformation. It has been proposed that this proton transits through a C-terminal pathway[25] that carries a passive inward $H^+$ leak when wild-type NKAs are in non-saturating concentrations of extracellular $Na^+$ and $K^+$ [42]. This leak has a complex dependence on extracellular $Na^+$, being inhibited at physiological concentrations, but activated at $Na^+$ concentrations ≤5 mM[43]. Therefore, provided the proton permeation pathway through site III is maintained, various NKA

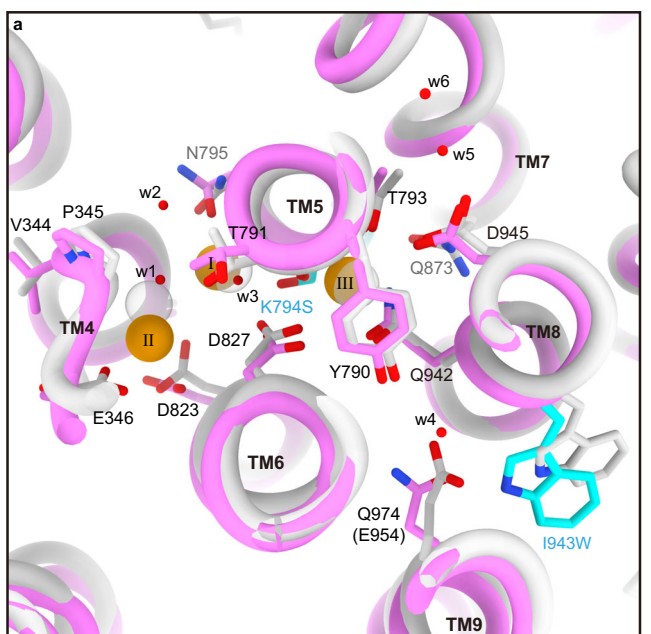
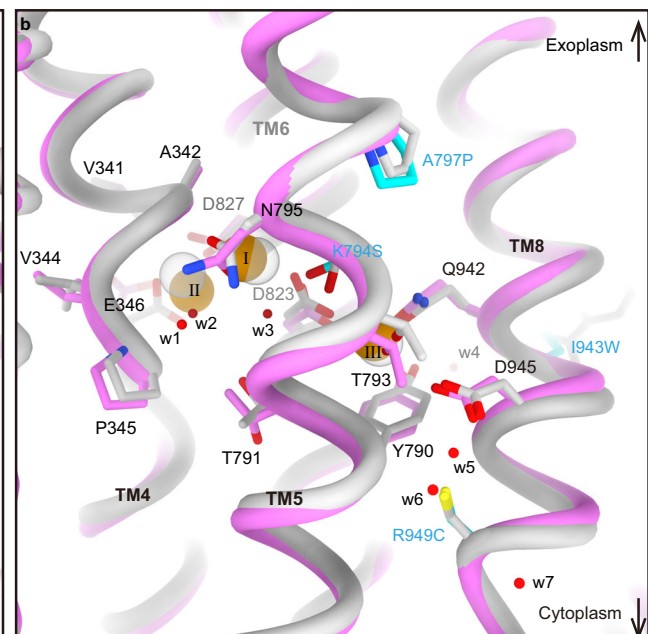

**Fig. 7 | Comparison of the cation-binding site between SPWC-ngHKA and NKA.**
**a, b** Overlapped cation-binding sites of SPWC-ngHKA in the 3Na⁺·E1·ATP state (pink tubes and sticks) and the (3Na⁺)E1P·ADP state of NKA (light gray, 3wgu) viewed from cytoplasmic side (**a**), or parallel to the membrane with the extracellular side-up (**b**). Mutated residues are indicated with cyan carbons. The three Na⁺ ions (orange) and seven water molecules (w1–w7, red) identified in the cryo-EM map of SPWC-ngHKA are shown as spheres (Fig. 6). All models are aligned by their immobile TM7-10 region.

mutations that reduce affinity for Na⁺ frequently reduce the ion's leak-inhibitory effect and show enhanced leak currents in the presence of physiological Na⁺ concentrations[27,29,30]. However, mutations that directly interrupt the H⁺ pathway at site III block the leak[44,45]. The NKA-mimicking SPWC mutant has an approximately threefold reduced apparent affinity for extracellular Na⁺ than the NKA, based on the center of the Q–V curve (Fig. 5e), something that may reflect an E2P-poised conformational preference that reduces apparent affinity for Na⁺ instead of an effect on the real affinity of the ion-binding sites. This reduced apparent affinity for Na⁺ can explain the leak currents seen in the presence of 125 mM Na⁺ (Fig. 5b). Interestingly, our Na⁺-bound E1 structure of the SPWC mutant has a water-filled cavity that reaches Asp945 and is capped by the α-subunit C-terminal tyrosine residues in the intracellular side (Fig. 10). We speculate that this cavity provides the pathway for H⁺ to tune Na⁺-binding and for the large leaks observed in the SWC and SPC mutants (Fig. 4a–c) and for the smaller leak current in the SPWC mutant (Fig. 5a, b).

The other two Na⁺ ions bind to sites I and II. The effect of the quadruple mutation at these binding sites is unexpected because, except for Lys794Ser, which is located between sites III and I, most introduced mutations are rather close to site III (Fig. 6e). Even though Ala797Pro does not contribute to ion coordination, the introduced proline appears to introduce a kink in TM5, right where the three sites must be formed[8]. This probably helps to position the side chains of the serine introduced at 794 (contributing to the coordination of site I and site III) and Asn795 (contributing to the coordination of site I, and indirectly to site II via a water molecule) to coordinate and stabilize the two Na⁺ ions. Furthermore, the proline side chain is also near the side chain of Gln873, another site III-coordinating residue in TM7 (Figs. 7a and 8f, g). These observations help explain how the E1 conformation of the SPWC mutant becomes compulsorily Na⁺ selective (Fig. 5f).

Our experiments help to definitely identify the determinants of H⁺/Na⁺ selectivity by these related pumps. Mutation of lysine 794 to serine or alanine reduces the ion selectivity of ngHKA, as both mutants can be activated by either Na⁺ or H⁺ (Fig. 3a), but neither single substitution suffices to flip selectivity. On the other hand, the NKA (which naturally has a serine where HKAs have a lysine) becomes electroneutral but remains strictly Na⁺ dependent when the Cys930Arg mutation was introduced to mimic Arg949 in ngHKA (cf. Fig. 3B in ref. 33). Therefore, the appearance of strict Na⁺selectivity requires addition of the trypto-phan to form site III (although Asp926 remains neutralized in NKA's Cys930Arg mutant). Taken together the findings described above indi-cate that strict H⁺ selectivity requires the simultaneous destruction of site III (by introducing Ile943 and Arg949, making the pump non-selective) and site I (by the tethered cation K794); a prediction to be tested in the future.

In summary, we presented the first structure of the ngHKA, an important drug target member of the P-type ATPase family, defined its stoichiometry and proton extrusion mechanism, which is distinct from the gastric pump, and revealed the requirements to transform this strict electroneutral 1H⁺:1K⁺ primary-active transporter into a strict electrogenic 3Na⁺:2K⁺ one. This transformation reveals how evolution exploited the same mechanism to build electrochemical gradients of different magnitude for dissimilar ions. Similar mechanisms may be involved in transitioning from H⁺-exclusive to Na⁺-exclusive ion trans-port in secondary-active transporters that have switched from H⁺-dependent to Na⁺-dependent transport throughout evolution[1,2]. Even if mutagenesis has been able to tune the selectivity of a Na⁺/Li⁺/H⁺ site capable of driving uphill sugar transport[46] we are unaware of other studies performing a rational transformation of a transporter that exclusively transports H⁺ into one that exclusively transports Na⁺. Although structural divergence hinders the extrapolation of our find-ings to those transporters, our results will likely lead to similar studies in those transporters.

## Methods

### Oocyte preparation and molecular biology for electrophysiology

Oocytes were enzymatically isolated, as previously described[36,47], and maintained at 16 °C in SOS solution (100 mM NaCl, 2 mM KCl, 1.8 mM CaCl₂, 1 mM MgCl₂, and 5 mM HEPES (pH 7.5 with NaOH), supple-mented with horse serum, antibiotic-antimycotic solution (Gibco) and

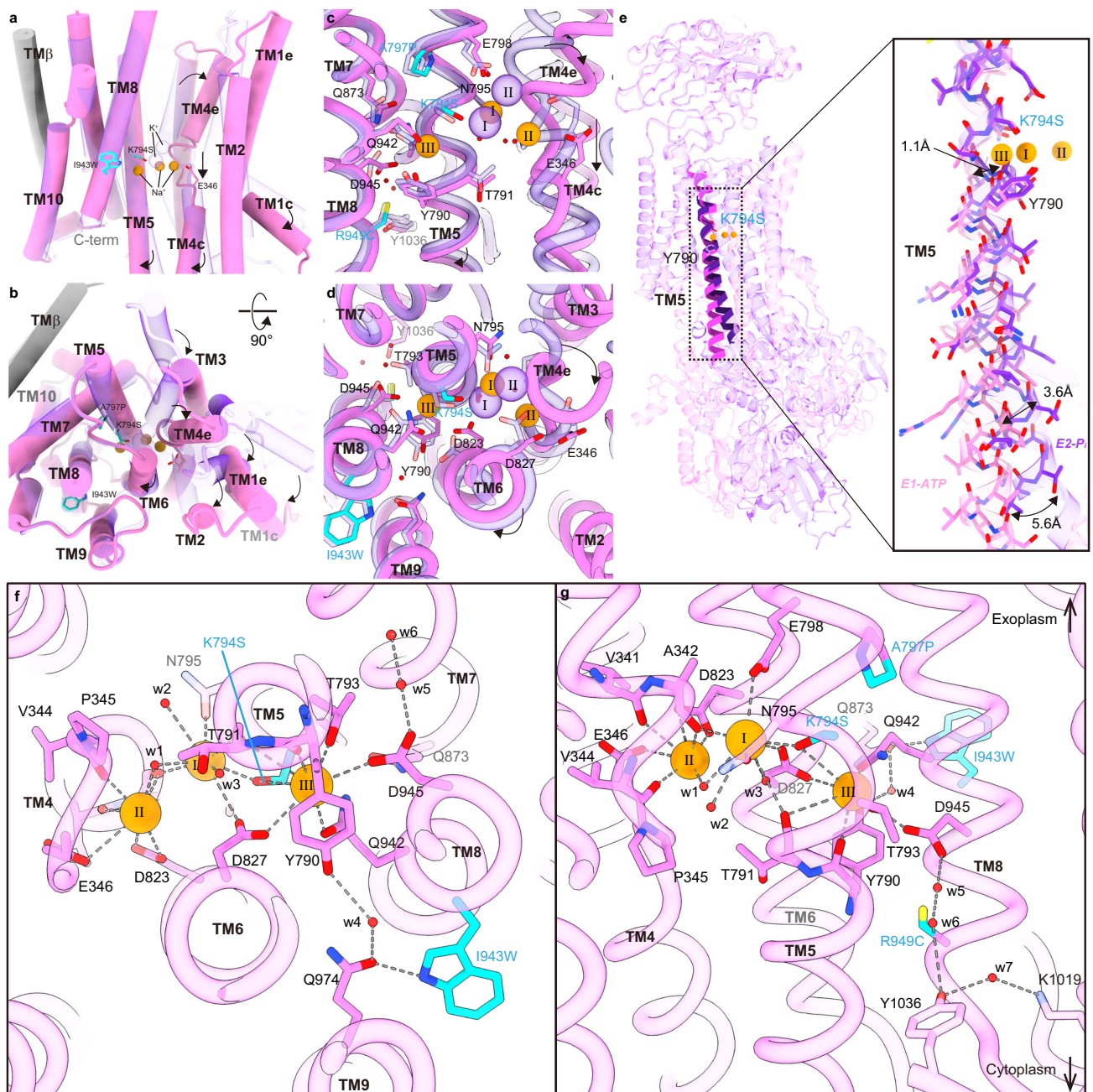

**Fig. 8 | Na⁺-coordination and conformational changes of the SPWC-ngHKA mutant. a–d** Rearrangement of TM helices (**a**, **c**: side view, **b**, **d**: extracellular view) of the whole TM region (**a**, **b**) and around the cation-binding sites (**c**, **d**). Pink and transparent purple models represent 3Na⁺·E1-ATP and (2 K⁺)E2-P_i state, respectively. Arrows indicate the displacement of indicated TM helices from (2 K⁺)E2-P_i to 3Na⁺·E1-ATP state. **e** Bent (purple, (2K⁺)E2-P_i) and straightened (pink, 3Na⁺·E1-ATP) TM5 helix. Overall view (left), and close-up view (right). Arrows indicate displacements observed between the two states. **f, g** Cation-binding sites in 3Na⁺·E1-ATP, viewed from cytoplasmic side (**f**) or from the membrane plane with exoplasmic-side-up (**g**). Orange spheres are the Na⁺ ions at cation-binding sites I, II, and III (see also Fig. 7 for comparison with NKA). Dashed lines indicate polar interactions at ~4 Å. The four mutations are represented with cyan carbons. For clarity, TM6 was removed from the model in panels **a**, **c**, **e**, and **g**.

gentamicin (Sigma)). Following injections with fifty nL of a cRNA mixture containing 1–2 μg/μL of α-cRNA and an equimolar amount of β-cRNA the oocytes were kept for up to a week in SOS solution. For NKA experiments, the injected cRNA was in vitro transcribed from cDNA of Xenopus Na⁺/K⁺ pump isoform with the ouabain-resistant double substitution Q119R/N131D α1-mutant and the β3 subunit. For ngHKA experiments, the injected cRNA was in vitro transcribed from cDNA of the rat non-gastric H⁺/K⁺ pump isoform (a generous gift from the late Käthi Geering) with an equimolar concentration of NKA β1-cRNA. Sequencing confirmed that this ngHKA clone had Gly315 as

described below in the protein expression methods. The ngHKA has very low ouabain sensitivity[23]. The reduced ouabain affinity of the templates used here allows us to selectively inhibit the endogenous oocyte pumps by preincubation with 10 μM ouabain, enabling exclusive measurement of signals from exogenous pumps[47].

## Electrophysiology

Before recording, oocytes were either Na⁺-loaded, to saturate intracellular-facing Na⁺-binding sites, or Na⁺-depleted (to remove most intracellular Na⁺). Na⁺ depletion was done in either a K⁺-loading or

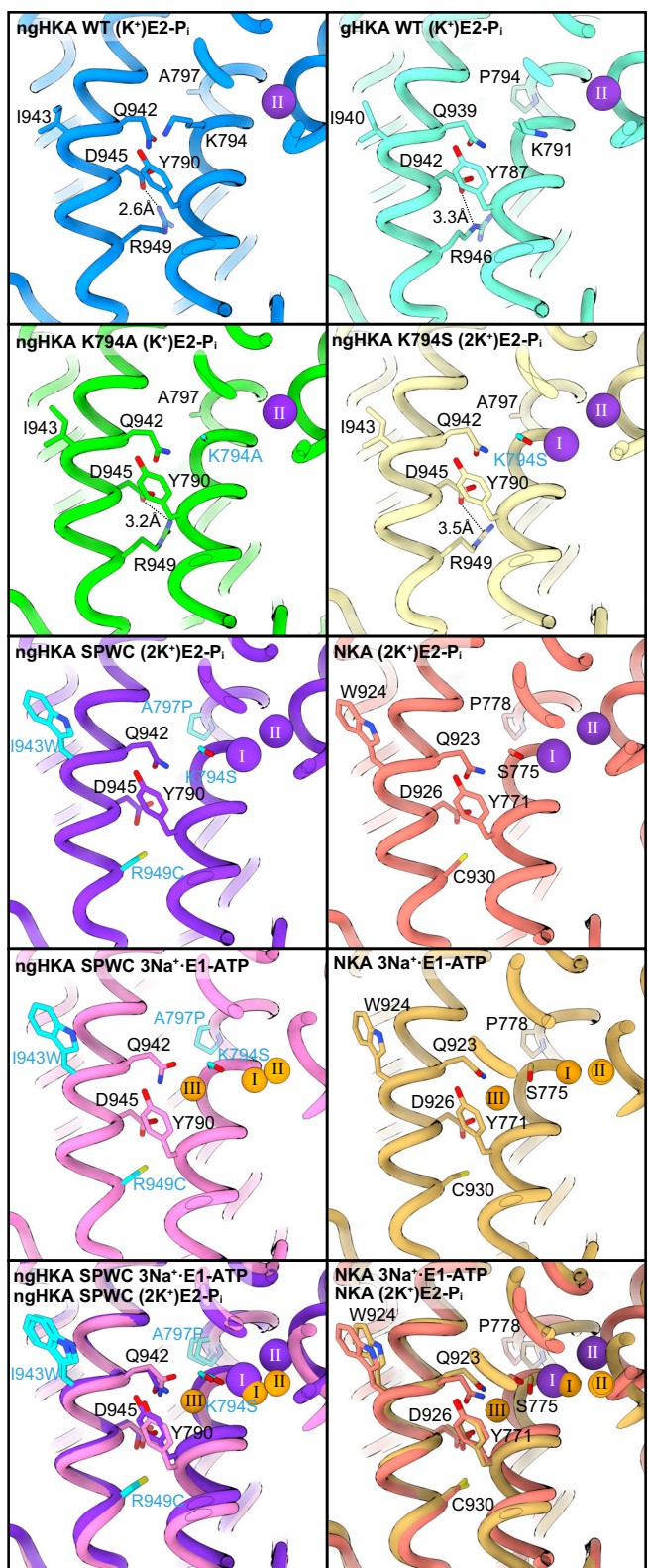

**Fig. 9 | A salt bridge between Asp945 and Arg949 of ngHKA.** Close-up views of the TM5 and TM8 region where the Na$^+$-binding-site-III-coordinating Asp945 is located in SPWC-ngHKA. The equivalent positions for gHKA and NKA are shown. Asp945 forms a salt bridge (dotted lines) with Arg949 in the E2-P$_i$ structures of WT-, K794A-, and K794S-ngHKA. This salt bridge is also observed in gHKA structures, while obviously absent in SPWC-ngHKA and NKA, due to the replacement of Arg949 with Cys. Comparison of 3Na$^+$·E1-ATP and (2K$^+$)E2-P$_i$ state of SPWC-ngHKA and its equivalent states for NKA (bottom panels) show little displacement in this site-III region during the transport cycle.

an NMG$^+$-loading solution. These procedures were done by a 1-h incubation in a solution containing (in mM) 90 cation [either NaOH (Na$^+$-loading), KOH (K$^+$-loading), or NMG$^+$ (NMG$^+$-loading)], 20 tetra-ethylammonium-OH, 0.2 EGTA, and 40 HEPES (pH 7.2 with sulfamic acid), supplemented with 10 μM ouabain. The extracellular solution contained (also in mM): 133 methane-sulfonic acid (MS), 10 HEPES, 5 Ba(OH)$_2$, 1 Mg(OH)$_2$, 0.5 Ca(OH)$_2$, 125 NaOH (Na$^+$ solution). External K$^+$ was added from a 450 mM K$^+$-MS stock. Ouabain was directly added to the extracellular solution. Its solubility above 10 mM was achieved by warming the solution & vortexing on the day of the experiment. These solutions were allowed to cool to room temperature before oocyte perfusion.

Two-electrode voltage clamp was performed at room tempera-ture (21–23 °C), with an OC-725C amplifier (Warner Instruments), a Digidata 1440 A/D board, a Minidigi 1A, and pClamp 10 software (Molecular Devices). Signals were filtered at 2 kHz and digitized at 10 kHz. Resistance of both microelectrodes (filled with 3M KCl) was 0.5–1 MΩ.

## Protein expression and purification for ATPase and structural studies

Procedures for protein expression are essentially the same as those reported previously[17], with some modifications. Briefly, a hexahistidine tag and the enhanced green fluorescence protein (EGFP) were inserted in the amino-terminal side of Met52 of the rat non-gastric HKA α-subunit and followed by a tobacco etch virus (TEV) protease recogni-tion sequence and subcloned into a hand-made vector[17]. It was pre-viously reported[14] that the database sequence UniPort ID: P54708 has an error Asp315 in TM3 ($^{311}$AVSIDIIFFI$^{320}$). We first expressed the data-base sequence and found a large K$^+$-independent activity (44.5 μmol/mg/h, $n = 1$) that was inhibited by the presence of K$^+$ with IC$_{50}$ = 18 mM. Given the result of that initial experiment, we corrected the sequence by introducing Asp315Gly as suggested in reference 14. The wild-type rat NKA β1 subunit was also cloned with the Flag epitope tag (DYKDDDDK) and the TEV protease recognition site in its N-terminus. The αβ-complex of ngHKA was expressed in the plasma membrane using baculovirus-mediated transduction of mammalian HEK293S GnT1$^-$ cells (BacMam) purchased from ATCC[48]. The harvested cells were broken up using a high-pressure emulsifier, and membrane fractions were sedimented.

For crystallization, membrane fractions were solubilized with 1% octaethylene glycol monododecyl ether (C$_{12}$E$_8$, Nikko Chemical) with 40 mM MES/Tris (pH 6.5), 10% glycerol, 5 mM dithiothreitol in the presence of 100 mM KCl, 1 mM MgCl$_2$, 1 mM AlCl$_3$, 4 mM NaF (to form the (K$^+$)E2-P$_i$ state), on ice for 20 min. Proteins were affinity purified by anti-Flag M2 affinity resin (Sigma), which was followed by digestion of affinity tag and deglycosylation by TEV protease and His-tagged endoglycosidase, respectively, at 4 °C overnight. Samples were further purified by a size-exclusion column chromatograph (SEC) using a Superose6 Increase column (Cytiva). Peak fractions were collected and concentrated to 10 mg/ml. The concentrated ngHKA samples were added to glass tubes in which a layer of dried dioleoyl phosphati-dylcholine had formed, in a lipid-to-protein ratio of 0.1–0.4, and incubated overnight at 4 °C in a shaking mixer operated at 120 rpm[49]. After removal of insoluble material by ultracentrifugation, the lipi-dated samples were used for crystallization.

For cryo-EM analysis, cells expressing ngHKA constructs were directly solubilized with 1% lauryl maltose neopentyl glycol (LMNG) in the presence of 40 mM MES/Tris (pH 6.5), 10% glycerol, 5 mM dithio-threitol, 1 mM MgCl$_2$, in the presence of 100 mM KCl, 1 mM AlCl$_3$, 4 mM NaF for (K$^+$)E2-P$_i$ state, or in the presence of 200 mM NaCl for 3Na$^+$·E1-ATP state, on ice for 20 min. After removing insoluble material by ultracentrifugation, the supernatant was mixed with anti-GFP nano-body resin[50] at 4 °C for 2 h, which was followed by washing with buffer containing 40 mM MES/Tris (pH 6.5), 5% glycerol, 1 mM MgCl$_2$, and

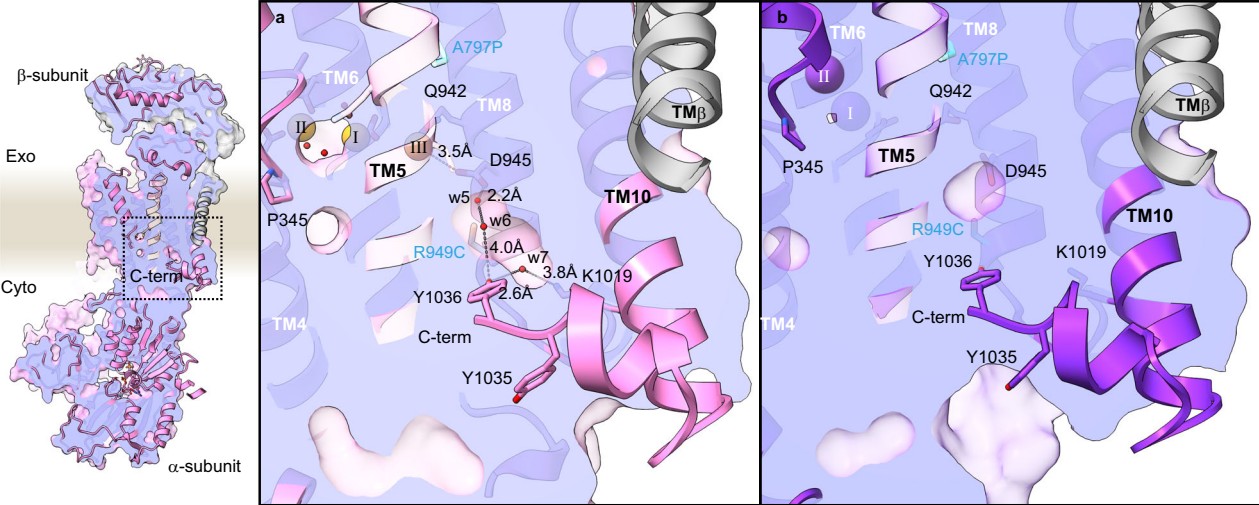

**Fig. 10 | C-terminal water-filled cavity.** Clipped membrane slice of C-terminal region of SPWC-ngHKA in the 3Na⁺·E1-ATP (**a**) and (2K⁺)E2-Pᵢ states (**b**) at the position indicated by a black box on the zoomed-out view of the molecule, viewed from a plane approximately parallel to the membrane, with extracellular side-up. Orange, purple, and small red spheres indicate Na⁺, K⁺, and waters, respectively. Dotted lines show potential hydrogen bonds (distance indicated) forming a network reaching the cation-binding site III from the C-terminus which is capped by the C-terminal tyrosine residues.

0.06% glycerol-diosgenin (GDN), in the presence of 100 mM KCl, 1 mM AlCl₃, 4 mM NaF to form the (K⁺)E2-Pᵢ state, or 200 mM NaCl to form the 3Na⁺·E1 state. After the addition of TEV protease and endoglycosidase, anti-GFP nanobody was incubated at 4 °C overnight. Digested peptide fragments containing EGFP and endoglycosidase were removed by passing the fractions through a Ni-NTA resin (Qiagen). Flow-through fractions were concentrated and subjected to a size-exclusion column chromatograph using a Superrose6 Increase column equilibrated in buffer comprising 20 mM MES/Tris (pH 6.5), 1% glycerol, 5 mM MgCl₂, and 0.06% GDN with 100 mM KCl, 1 mM AlCl₃, 4 mM NaF for (K⁺)E2-Pᵢ state, or 300 mM NaCl for 3Na⁺·E1-ATP state. Peak fractions were collected and concentrated to 8 mg/ml. The final concentration of 5 mM AMPPCP was added to the protein sample for 3Na⁺·E1-ATP state.

### Crystallization and structural determination

Crystals were obtained by vapor diffusion at 20 °C. A 5-mg/ml purified, lipidated protein sample was mixed with reservoir solution containing 10% glycerol, 33% PEG300, 50 mM Glycine-NaOH pH 9.5, 100 mM NaCl, 200 mM KCl, 0.1 mM AlCl₃, 0.4 mM NaF. Crystals were flash frozen in liquid nitrogen. For the Rb⁺-bound crystal, RbCl substituted KCl.

Diffraction data were collected at the SPring-8 beamline BL41XU and BL45XU, and processed using Kamo[51] and XDS[52]. Structure factors were subjected to anisotropy correction using the UCLA MBI Diffraction Anisotropy server[53] (http://services.mbi.ucla.edu/anisoscale/). The structures were determined by molecular replacement with PHASER[54], using an atomic model of gastric H⁺,K⁺-ATPase in K⁺-occluded E2-Pᵢ form (pdb ID: 6jxh) as a search model. Coot (0.9.2)[55] was used for cycles of iterative model building, and Refmac5[56] and Phenix (1.18)[57] were used for refinement. Rubidium ions were identified in anomalous difference Fourier maps calculated using data collected at wavelengths of 0.8147 Å. The model contained 92.0/8.0/0.0% in the favored, allowed, and outlier regions of the Ramachandran plot.

### Cryo-EM analysis

Preparation of sample and cryo-EM grid was done according to a previous report[58]. The purified protein samples (at 8 mg/ml) were applied to a freshly glow-discharged Quantifoil holey carbon grid (R1.2/1.3, Cu/Rh, 300 mesh), using a Vitrobot Mark IV (FEI) at 4 °C with a blotting time of 8 s under 99% humidity, and the grids were then plunge-frozen in liquid ethane. The prepared grids were transferred to a Titan Krios G4i microscope (Thermo Fisher Scientific), running at 300 kV and equipped with a Gatan Quantum-LS Energy Filter (GIF) and a Gatan K3 Summit direct electron detector in the electron counting mode. Imaging was performed at a nominal magnification of ×105,000, corresponding to a calibrated pixel size of 0.83 Å/pix (The University of Tokyo, Japan). Each movie was recorded in a correlated-double sampling (CDS) mode for 5 s and subdivided into 50 frames. The electron flux was set to 7.3 e⁻/pix/s at the detector, resulting in an accumulated exposure of 53 e⁻/Å² at the specimen. The data were automatically acquired by the image shift method using SerialEM (7.8) software[59], with a defocus range of −0.8 to −1.6 μm. The dose-fractionated movies were subjected to beam-induced motion correction, using MotionCor2[60] or Relion (3.2)[61], and the contrast transfer function (CTF) parameters were estimated using CTFFIND4[62].

For each dataset, particles were initially picked by using EMAN2.2[63], and extracted with downsampling to a pixel size of 3.24 Å/pix. These particles were subjected to several rounds of 2D and 3D classifications. The best class was then re-extracted with a pixel size of 0.83 Å/pix and subjected to 3D refinement. The resulting 3D model and particle set were subjected to per-particle defocus refinement, beam-tilt refinement, Bayesian polishing[64], and 3D refinement. The resolution of the analyzed map was defined according to the FCS = 0.143 criterion (Supplementary Fig. 3)[65]. The local resolution and angular distributions for each structure were estimated by Relion. All the models were manually built in Coot using the model derived from the crystal structure of ngHKA WT. For the SPWC E1-ATP state, due to the weak EM density for A and N-domains, each domain was fit into the density map as a rigid body and was not further refined. Phenix (ver 1.19)[57] was used for the refinement of other regions. The E2-Pᵢ state of K794A-, K794S-, SPWC-ngHKA, and E1-ATP state of SPWC-ngHKA models contained 96.0/4.0/0.0%, 96.5/3.5/0.0%, 96.0/4.0/0.0%, and 97.2/2.8/0.0% in the favored, allowed, and outlier regions of the Ramachandran plot, respectively.

### ATPase activity in membrane fractions

The N-terminal GFP-tagged, N-terminal deleted (Δ52) ngHKA α subunit used for structural analysis were co-expressed with the wild-type NKA β1 subunit using the BacMam system as described above, and broken membrane fractions were collected. For the activity measurement of gHKA and NKA, similar N-terminal tagged α-subunits were co-expressed with their accessory subunits (pig gHKA α-subunit (ATP4A) with pig

HKA β-subunit (ATP4B), and human NKA α1 subunit (ATP1A1) with human β3 subunit (ATP1B3) and human FXYD5-subunit, respectively). ATPase activity was measured as described previously[17]. Briefly, permeabilized membrane fractions were suspended in buffer containing 40 mM PIPES/Tris (pH 7.0), 2 mM MgCl$_2$, 2 mM ATP di-tris salt, 0–100 mM KCl and/or 0–300 mM NaCl in the presence of 1 μM ouabain (to inhibit endogenous NKA) and thapsigargin (to inhibit SERCA) in 96-well microtubes. We also measured ATPase activity in the presence of 1 mM BeSO$_4$ and 3 mM NaF. This forms a BeF complex which acts as an irreversible and non-specific ATPase inhibitor, to estimate K$^+$-and Na$^+$-independent ATPase fractions. Reactions were initiated by incubating the fractions at 37 °C using a thermal cycler and maintained for 1 h. Reactions were terminated by adding 2 M HCl, and the amount of released inorganic phosphate was determined colorimetrically[66] using a microplate reader (TECAN). The specific K$^+$-dependent ATPase activity was calculated by subtracting the activities in the absence of K$^+$ and Na$^+$, giving the maximum K$^+$-sensitive specific activities of 4.1 μmol/mg/h for WT-ngHKA at 80 mM KCl, 24.7 μmol/mg/h for K794A-ngHKA at 80 mM KCl and 300 mM NaCl, 12.8 μmol/mg/h for K794S-ngHKA at 5 mM KCl and 300 mM NaCl, 2.7 μmol/mg/h for gHKA at 30 mM KCl, 1.0 μmol/mg/h for SPWC-ngHKA at 5 mM KCl and 300 mM NaCl and 7.8 μmol/mg/h for NKA at 100 mM KCl and 300 mM NaCl. When BeF$_3$-inhibited samples were used as background, activity in the absence of K$^+$ and Na$^+$ was 1.8 μmol/mg/h for WT-ngHKA, 4.4 μmol/mg/h for K794A-ngHKA, 2.7 μmol/mg/h for K794S-ngHKA, 0.54 μmol/mg/h for gHKA, 0.88 μmol/mg/h for SPWC-ngHKA, 0.30 μmol/mg/h for NKA, and 0.24 μmol/mg/h for mock-transfected cells. For Fig. 3a, the maximum measured K$^+$-activated activities without Na$^+$ (set as 100% activity) were 4.1 μmol/mg/h for WT at 80 mM KCl, 15.6 μmol/mg/h for K794A at 100 mM KCl and 7.2 μmol/mg/h for K794S at 20 mM KCl. The half-maximal activating K$^+$ concentrations (K$_{0.5,K+}$, from Hill equation fits to the data, Eq. 3) were K$_{0.5,K+}$ = 76 ± 0.87 μM ($nH$ = 0.93 ± 0.27) for WT, K$_{0.5,K+}$ = 280 ± 86 μM ($nH$ = 1.36 ± 0.16) for Lys794Ser and K$_{0.5,K+}$ = 200 ± 47 μM ($nH$ = 0.97 ± 0.15) for Lys794Ala (mean ± SD, $n$ = 3). The K$_{0.5}$ in the presence of 100 mM Na$^+$ was K$_{0.5,K+}$ = 498 ± 130 μM, $nH$ = 1.1 ± 0.07 for WT, K$_{0.5,K+}$ = 528 ± 14 μM, $nH$ = 1.47 ± 0.10 for Lys794Ser and K$_{0.5,K+}$ = 3900 ± 1500 μM, $nH$ = 0.90 ± 0.09 for Lys794Ala.

The membrane fractions (1 mg/ml) used for the ATPase measurement were solubilized with 1% LMNG in the presence of 1 mM BeSO$_4$, 3 mM NaF, 1 mM MgCl$_2$, 1% glycerol, and 40 mM MES/Tris, pH 6.5 for 20 min on ice. After removing insoluble materials by ultracentrifugation, the supernatant was analyzed by fluorescence size-exclusion chromatography (FSEC) monitoring GFP fluorescence (Ex 495 nm, Em 520 nm). The peak fluorescence values (in the arbitrary unit, a.u.) serve as a measure of the relative amount of enzyme in the membrane fraction. Compared to NKA (peak value 1.5 a.u.) and gHKA (1.6 a.u.), ngHKA WT (4.6 a.u.) and its single mutants, K794A (26.6 a.u.) and K794S (14.7 a.u.) show higher expression level, consistent with their relatively high specific activities in the membrane fraction. The SPWC mutant, in contrast, had a similar expression to NKA (1.8).

### Valence calculation

The valence (v) for the specific cation (M$^+$) was calculated using the equation[17]

$$v_{M^+} = \sum_{j=1}^{m} v_j = \sum_{j=1}^{m} \left( \frac{R_j}{R_0} \right)^{-N} \tag{1}$$

where $v_j$ is the partial valence contributed by the $j$th ligating oxygen in the coordination shell located at a distance $R_j$, and $m$ is the total number of oxygen atoms within 4.0 Å. The parameters $R_0$ (1.622 for Na$^+$ and 2.276 for K$^+$) and N (4.29 for Na$^+$ and 9.1 for K$^+$) translate the bond length into the bond strength, or valence, and are specific for a given metal ion-oxygen pair[67].

### Electrophysiology data analysis

The amount of charge transferred was calculated by integrating the transient current when the voltage pulse was turned off (Q$_{OFF}$). For mutants with a sigmoidal voltage dependence, the charge vs. voltage (Q–V) curve was fitted with a Boltzmann distribution:[42]

$$Q_{OFF} = Q_{hyp} + \frac{Q_{tot}}{1 + \exp\left( \frac{z_q e (V - V_{1/2})}{kT} \right)} \tag{2}$$

where $Q_{hyp}$ is the charge moved by hyperpolarizing pulses, $Q_{tot}$ (|$Q_{hyp}$| from negative pulses + |$Q_{pos}$| from positive pulses) is the total charge moving over the whole voltage axis, $V_{1/2}$ is the center of the distribution, $z_q$ is the valence of the moving charged particle if it crossed the entire membrane electric field, $e$ is the elementary charge, $k$ is the Boltzmann constant, and $T$ is the absolute temperature. $kT/zqe$ is also referred to as the slope factor (the smaller $kT/z_q e$ the steeper the slope).

The ligand concentration dependencies of the distinct enzymatic reactions and currents were fitted by a Hill function:

$$A = A_0 + A_{max} \cdot \left( \frac{[L]^{nH}}{K_{0.5}^{nH} + [L]^{nH}} \right) \tag{3}$$

Where $A_O$ is the current or ATPase activity in the absence of the ligand, $A_{max}$ is the current/activity at infinite ligand concentration, $K_{0.5}$ is the ligand concentration giving half-maximal activation and $nH$ is the Hill coefficient. Whenever inhibition was observed at high K$^+$ concentrations, a negative Hill function was added to the function in Eq. (3).

All electrophysiological data analysis was performed with pClamp and Origin (OriginLab), and ATPase analysis with GraphPad Prism9. As indicated, error bars presented in graphic form are SEM. Errors in the main text are SD (unless otherwise stated).

### Reporting summary

Further information on research design is available in the Nature Research Reporting Summary linked to this article.

## Data availability

The structural data generated in this study have been deposited in the Protein Data Bank and EM Data Bank under accession codes 7X20: Crystal structure of non-gastric H,K-ATPase alpha2 in (K$^+$)E2-AlF state, 7X21 and EMD-32954: Cryo-EM structure of non-gastric H,K-ATPase alpha2 K794A in (K$^+$)E2-AlF state, 7X22 and EMD-32955: Cryo-EM structure of non-gastric H,K-ATPase alpha2 K794S in (2K$^+$)E2-AlF state, 7X23 and EMD-32956: Cryo-EM structure of non-gastric H,K-ATPase alpha2 SPWC mutant in 3Na$^+$E1-AMPPCP state, and 7X24 and EMD-32957: Cryo-EM structure of non-gastric H,K-ATPase alpha2 SPWC mutant in (2K$^+$)E2-AlF state. Source data are provided with this paper.

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

## Acknowledgements

The synchrotron radiation experiments were performed at BL41XU and BL45XU in SPring-8 with the approval of the Japan Synchrotron Radiation Research Institute (JASRI Proposal numbers: 2019B2707 and 2021B2716). Cryo-EM experiments were supported by the Platform project for Supporting Drug Discovery and Life Science Research [Basis for Supporting Innovative Drug Discovery and Life Science Research (BINDS)] from AMED under Grant Number JP19am0101115j0003 (support number 1925). We thank beamline and cryo-EM staff for their facilities and support; Ms. Y. Dou and Dr. J. Suzuki for providing the NKA clones used in this study; Dr. K. Hayashida for helping cryo-EM grid screening; and Dr. Craig Gatto for comments on the manuscript. This work has been funded by grants from the National Science Foundation (MCB-2003251) and CH Foundation to P.A., and from Grant-in-Aid for Scientific Research (21H02426), Takeda Science Foundation, Uehara Science Foundation, Naito Foundation, ONO Medical Research Foundation, Novartis Foundation, and BINDS from AMED (JP21am010074) to K.A.

## Author contributions

P.A. and K.A. conceptualization; H.N. and K.A. performed protein purification, activity measurement, crystallization, and crystal data collection; H.N., A.O., and K.A. cryo-EM grid preparation; T.N. and K.A. cryo-EM data acquisition; K.A. crystallographic and cryo-EM analysis; V.Y., D.M., and P.A. electrophysiological analysis; V.Y., P.A., and K.A. writing; A.O., P.A., and K.A. funding acquisition; P.A. and K.A. project administration.

## Competing interests

The authors declare no competing interests.
