## [Peer Review File · Nature Communications]

Structure and function of H⁺/K⁺ pump mutants reveal Na⁺/K⁺ pump mechanismsREVIEWERS' COMMENTS

Reviewer #1 (Remarks to the Author):

This interdisciplinary paper uses a number of rigorous approaches to determine what it takes for a membrane protein to adapt its function. The P-type ATPases evolved to break all the rules and drive the uphill transport of their substrates across biological membranes. These investigators broke new ground by determining the structures of nongastric H,K-ATPase in two opposing conformations (using both crystallography and cryo-EM), and determined the missing conformation for the gastric H,K-ATPase as well. The two ATPases have different roles in different tissues, and the new biochemical and electrophysiological evidence goes a long way to explain how and why.

The structures give new insight into differences in the arrangement of amino acids in the ion binding site, which is buried in the membrane where it has alternating access to the two sides of the membrane. The structures suggested hypotheses that were tested by a series of mutations that swapped unique amino acids. Function was then analyzed by rigorous electrophysiological methods that measure either pump current (ion transfer to the other side) or current transients (ion movements as partial reactions) under conditions where one of the substrates was missing (K⁺, Na⁺); over a wide range of transmembrane voltages; and determined if the resulting movement of ions was electrogenic or not, a key functional difference. The kinetics of ATP hydrolysis were also analyzed.

The paper presents a series of hypotheses where the electrophysiological evidence is explained first, and then the insights gained from the structures. In the end, the authors succeeded in producing a group of mutations that functionally changed the nongastric pump into a Na,K-ATPase, with all of its key features and with structural verification of the predicted ion occupancies. The interpretation of the structural features is interesting and well-supported.

Critique:

There is no criticism of the approach, the findings, or the writing. The specific conclusions seem accurate and convincing.

Some of the figures were hard to read because the quality of the PDF conversion was quite poor (possibly the journal's fault?). Authors should review and reject a conversion if figure quality is this distorted. When making your own PDFs, be sure to select high resolution.

1. Can the authors refer to evidence that *Xenopus* Na,K-ATPase (functions best at low temperature) instead of rat Na,K-ATPase (functions best at 37°C) can be compared to rat Na,K-ATPase when both are investigated at low temperature?

2. It would be helpful to have a side-view overview structure illustration of the mutations in the main manuscript.

3. The movies were not helpful at all, except for the final one. The others rotate too fast and give the viewer nothing to use for orientation. Please redo them in the style Toyoshima uses, or omit them.

The last movie would benefit from some labeling.

Minor:

Why are the reference numbers in text in different formats?

Inappropriate commas: Line 23, 53 after III,

65: missing parenthesis

172-173: Would be better without a paragraph break or starting the new paragraph with "To confirm..."

336: suffices

355: transports

509-510: no need to say anything about ouabain stock solubility.

520: I don't understand. If uniprot P54708 has a sequencing error, how did the aspartate get into the clone you were using, necessitating correction by mutation?

728-730: use semicolons

Reviewer #2 (Remarks to the Author):

The manuscript by Young et al., is an interesting study on the mechanics of a H⁺/K⁺ pump especially in the engineering into a Na⁺/K⁺ pump. The combination of structural and biophysical approaches makes this a robust study and there should be some broad interest in the transport field. I had a few comments/suggestions on the manuscript which I hope will improve the clarity/robustness.

1) The current format seems to reflect that of another manuscript with the introduction blending into the results sections. It would improve clarity if these were more clearly defined and introduction expanded which at the moment is only 26 lines long.

2) Line 76 there is one open bracket but this is never closed in the text.

- 3) Line 101 the authors discuss electroneutrality and how a Lys794Ala mutation removes this with Glu doing this to a lesser extent. This could be expanded on or referred back to on the other mutations as it seems strange that changing to the opposite charge has only a modest effect here.
- 4) When conducting the ATPase activity assays in the membrane it is unclear how differences in expression level were accounted for. Was a western blot conducted on each membrane to show relative expression levels?
- 5) Figure 1 is very pixelated especially the alignment, I assume this is a conversion to pdf issue.
- 6) When looking at the validation reports all structures have worse clashscores than expected, this is most apparent for 7X20, is there a reason for this value being worse in this structure?
- 7) The complementary mutations to change the nature of the pump are interesting. Are there any examples in this family where nature has already made similar changes to repeat this?

Reviewer #3 (Remarks to the Author):

The manuscript is an important and exciting contribution that increases the understanding of ion selectivity in the binding sites of P-type ATPases and how the substitution of single amino acids may affect ion selectivity and stoichiometry of ion pumps. In summary, the replacement of only four amino acids of the (electroneutral) non-gastric H⁺/K⁺-ATPase close to the location of the classical sites I, II and III against the respective amino acids of the Na⁺/K⁺-ATPase transforms the protein into an electrogenic ion pump that exchanges 3 Na⁺ for 2 K⁺ across the membrane, as it is known from the Na⁺/K⁺-ATPase.

To approach this goal, the authors studied structure and function of a WT non-gastric H⁺/K⁺-ATPase and six mutants, which were purposefully chosen and constructed on the basis of previous knowledge of the proposed relevance of these amino acids with respect to ion binding. First, Lys794 was altered to alanine or serine, which introduced electrogenic ion transport, eliminated H⁺-exclusive dependence and altered Na⁺/K⁺ competition. The Lys794Ser showed larger transient currents in voltage-clamp experiments, and the cryo-EM structures of the K⁺-bound E2-P_i conformation displayed strong densities in sites I and II, indicating the presence of two K⁺. In the next step, the Arg949Cys mutation was studied, in which another seemingly important amino acid from the Na⁺/K⁺ pump was introduced to obtain the Lys794Ser/Arg949Cys construct. This mutant revealed no obvious progress, but the introduction of a third modification, Lys794Ser/Ile943Trp/Arg949Cys (SWC mutant), acknowledging the Trp of the Na⁺/K⁺-ATPase in the respective position turned out to be an advantageous step. The SWC mutant showed considerable outward current when exposed to K⁺, but seemed to cause an extreme reduction in the affinity for external Na⁺, and it required intracellular Na⁺ to act as ion pump. The authors tested also Lys794Ser/Ala797Pro/Arg949Cys mutant, in which another proposed crucial amino acid of the

Na⁺/K⁺ pump was introduced, which had, however, no promoting effect. But when the Ala797Pro was added to the SWC template (the SPWC mutant) a prototypical Na⁺/K⁺ pump was created. Electrophysiological experiments showed that the SPWC mutant mimicked the Na⁺/K⁺-ATPase rather well. The cryo-EM structures of both Na⁺/K⁺ pumps, especially in the membrane domain around the ion-binding sites, concurred amazingly well in both studies conformations, 3Na⁺·E1-ATP and (2K⁺)E2-P_i.

Considering the location of the four amino acids that transformed the H⁺/K⁺ pump into a “bona fide” Na⁺/K⁺ pump, it is surprising that the Ile943Trp mutation that is required to produce a functional site III interacts only by an allosteric influence, since the side chain of this amino acid is located on TM8 on the side opposite to site III. It contributes, however, with a hydrogen bond to Gln974 in TM9, thus contributing to a (larger) hydrogen bond network crucial for a functional site III.

The research concept presented in this manuscript is convincing and the applied methods are well established. The amount and quality of the experimental and analytical work are impressive, well performed and reliable. A wide spectrum of up-to-date methods were applied to obtain the presented results, comprising molecular biology, protein expression in oocytes and cell cultures, protein isolation, determination of ATPase activity, two-electrode voltage clamp experiments, protein crystallization and X-ray crystallography, cryo-EM analysis, and complex data analysis. All these techniques were performed professionally and reflect the expertise of the authors who are well established in their respective field by numerous first-class publications. The discussion is sound and does justice to the results. The extensive supplementary material supports to a better comprehension of the manuscript.

As one result of their work five new high-resolution protein structures were entered into the PDB (and four into the EMDB), among them the first structure of a non-gastric H⁺/K⁺-ATPase, of which the authors propose that it may become an important drug target member of the P-type ATPase family.

To further improve the comprehensibility of the paper this reviewer proposes to consider a number of minor corrections and modifications.

Minor comments:

I.42: “... a process promoting chronic respiratory ...”

I.58: Neither here nor in the methods it is explained why a rat NKA β subunit was used and instead of a nGHKA β subunit.

I.203: When 25 mV lead to a two-fold reduction, why does a (-85 mV) – (-51 mV) = (-34 mV) shift lead to a ~4-fold reduction? (Why not ~2.7-fold?)

I.249: “Compared to E2, this is accompanied...” What means “this”? The (overall) process?

I.264-268: When site III is occupied first (I.245), the displacement of 5.6 Å and P-domain movement will occur before sites I and II are occupied. But what prevents then an untimely autophosphorylation?

I.273: "... that the side chain of 943 faces ..."

I.303: "... Ile943, prevents Na⁺ binding ..."

I.356: "... extrapolation of our findings ..."

I.582: "... Å² ..."

I.633: "The amount of charge transferred was calculated by integrating ..."

I.752: "... (NKA, black) ..."

I.764: "... NKA crystal structure (wheat ..."

I.771: "... ngHKA WT (blue) ..."

I.772: "... maximum measured K⁺-activated ..."

I.781: Here and elsewhere (Figs. 2b, 3a, 3e, 4a, 4c) the length and position of the bars indicating application of substrates seem to be somewhat erratic in several places. Please replace by bars of correct length and position.

I.799: "... Lys794Ala (n = 7) and WT could not ..."

I.841: "... ouabain-sensitive (OS) current ..."

I.852: "... ATPase activity in ..."

I.854: "... (orange) mM Na⁺."

I.863: In case of 3Na⁺-E1-ATP (panel h) the β subunit is not shown in grey.

I.866: "... the 3 bound Na⁺ ..."

I.876: "... (e) bent (purple, ..."

I.881: "Dashed" instead of "dotted" lines?

I.920 "... activities of various ATPases studied"

Point-by-point Response

We thank the reviewers and the editor for the critical evaluation of our manuscript. We are delighted that all three reviewers find our findings important, interesting and expertly done. The critiques by the reviewers and in blue and our responses in black. In the manuscript a few sentences that were rewritten in response to their comments are marked blue. Changes to improve readability or to shrink the size of the figure legends to match the journal style were not marked.

Reviewer #1 Critique:

There is no criticism of the approach, the findings, or the writing. The specific conclusions seem accurate and convincing.

Some of the figures were hard to read because the quality of the PDF conversion was quite poor (possibly the journal's fault?). Authors should review and reject a conversion if figure quality is this distorted. When making your own PDFs, be sure to select high resolution.

1. Can the authors refer to evidence that *Xenopus* Na,K-ATPase (functions best at low temperature) instead of rat Na,K-ATPase (functions best at 37°C) can be compared to rat ngATPase when both are investigated at low temperature?

It is not completely clear to us what the reviewer means with the commentary that one type of protein functions best at one temperature. We don't think anyone has looked at the temperature dependence of ngHKA in detail and this goes beyond our goals for this manuscript. Regarding the *Xenopus* pumps, we looked at its function of the oocyte's endogenous pumps at two different temperatures (24 and 34°C cf. Fig. 5B Stanley et al 2015). Based on that, it doesn't look as the Q10 or the turnover rates at the two temperatures are different between mammalian and batrachian.

2. It would be helpful to have a side-view overview structure illustration of the mutations in the main manuscript.

Based on the journal recommendation we split Fig 4 into two figures (new Fig. 4 and Fig. 5) and added a new panel (Fig. 5e) indicating mutations in the overall structure and its close-up view as suggested by the reviewer.

3. The movies were not helpful at all, except for the final one. The others rotate too fast and give the viewer nothing to use for orientation. Please redo them in the style Toyoshima uses, or omit them.

The last movie would benefit from some labeling.

We omit Movies S1-5 in the new version. The old movie S6 is Movie S1 in the new manuscript.

Minor:

Why are the reference numbers in text in different formats?

We assume some issue with Endnote might have played a trick. We have tried to format them equally in the current version.

Inappropriate commas: Line 23, 53 after III,

Corrected

65: missing parenthesis

Parenthesis added

(**Fig. 1e**, PDB code 6jxh, the Tyr799Trp was used to stabilize the occluded conformation in reference¹¹)

172-173: Would be better without a paragraph break or starting the new paragraph with “To confirm...”

Changed as suggested.

336: suffices

Corrected.

355: transports

Corrected.

509-510: no need to say anything about ouabain stock solubility.

We removed the comment on lack of precipitation after dissolution. We believe it is important to mention how we get the high concentration in the final experimental (not stock) solution.

520: I don't understand. If uniprot P54708 has a sequencing error, how did the aspartate get into the clone you were using, necessitating correction by mutation?

We apologize for not making this much clearer. De Pont and colleagues (2005, JBC), had shown that Asp315 is the only database sequence with a Asp I lieu of the Gly (see figure).

They said that they assumed this is an error and simply worked with the Gly mutant. We had started by cloning the sequence of the database, before our structural work. Upon expression we found a large K⁺-independent activity (44.5 μmol/mg/h at pH 7.2) that was inhibited by K⁺ (*I*C₅₀ =18 mM). That activity was absent in non-transfected cells or in cells expressing any other clones. So, we also introduced the mutation that gave the normal K⁺-activated activity (D315G). Independently, for our electrophysiological studies we were working with a rat clone (given to us by the late Kathy Geering) that already had the G315 there. We think that it is important to mention this sequence error in the database as nobody could “repeat” the experiments if they used the wrong clone.

Substitution of G315D by sequencing error

J Biol Chem (2005) **280**, 33115-33122.

Rat		296	IAIEIEHFVHIVAGVAVS	DI	IIFFITAVCMKYYYV
XP_028643194.1		296	IAIEIEHFVHIVAGVAVS	IGI	IFFITAVCMKYYYV
XP_031218428.1		295	IAIEIEHFVHIVAGVAVS	IGI	IFFITAVCMKYYYV
XP_021058899.1		295	IAIEIEHFVHIVAGVAVS	IGI	IFFITAVCMKYYYV
XP_006993532.1		297	IAIEIEHFVHIVAGVAVS	IGI	IFFITAVCMKYYYV
XP_005372048.3		297	IAIEIEHFVHIVAGVAVS	IGI	IFFITAVCMKYYYV
XP_028713967.1		297	IAIEIEHFVHIVAGVAVS	IGI	IFFITAVCMKYYYV
ERE92206.1		321	IAIEIEHFVHIVAGVAVS	IGI	IFFITAVCMKYYYV
XP_021037267.1		295	IAIEIEHFVHIVAAVAVS	GV	IFFITAVCMKYYYV
NP_619593.2		295	IAIEIEHFVHIVAAVAVS	GV	IFFITAVCMKYYYV
AAL68709.1		295	IAIEIEHFVHIVAAVAVS	GV	IFFITAVCMKYYYV
XP_012972701.1		264	IAIEIEHFVHIVAGVAVS	IGV	WFFITAVCMKYYYV
XP_008822627.1		297	IAIEIEHFVHIVAGVAVS	IGI	IFFITAVCMKYYYV

Partial sequences of rodent HK α2 chain (amino acid identity is 93-97%)

We have now written this more clearly in the methods section:

It was previously reported ¹⁴ that the database sequence UniPort ID: P54708 has an error Asp315 in TM3 (³¹¹AVSIDIIFFI³²⁰). We first expressed the database sequence and found a large K⁺-independent activity (44.5 μmol/mg/h, n = 1) that was inhibited by the presence of K⁺ with *I*C₅₀ = 18 mM. Given the result of that initial experiment we corrected the sequence by introducing Asp315Gly as suggested in reference ¹⁴.

728-730: use semicolons

Corrected.

Reviewer #2 (Remarks to the Author)

1) The current format seems to reflect that of another manuscript with the introduction blending into the results sections. It would improve clarity if these were more clearly defined and introduction expanded which at the moment is only 26 lines long.

We appreciate the reviewer's comment, but we believe this format serves better for a general audience. We would like to know what specific change is requested, and would love the editor's input in case such a correction is considering necessary.

2) Line 76 there is one open bracket but this is never closed in the text.

Corrected (see response to reviewer 1)

3) Line 101 the authors discuss electroneutrality and how a Lys794Ala mutation removes this with Glu doing this to a lesser extent. This could be expanded on or referred back to on the other mutations as it seems strange that changing to the opposite charge has only a modest effect here.

It is beyond the scope of the manuscript to discuss the pioneering results of Horisberger beyond what we show here. We do not study Lys794Glu. Therefore, we feel it is not our place to speculate on why the Glu has electrogenicity, which is pretty minor compared to Ala.

4) When conducting the ATPase activity assays in the membrane it is unclear how differences in expression level were accounted for. Was a western blot conducted on each membrane to show relative expression levels?

The ATPase value in the membrane fraction is simply estimated as moles of Pi released / total proteins in the membrane / h. We did not pursue a turnover rate because it is hard to be solidly quantitative about it. We did not perform quantitative Western blotting to measure expression levels. Nevertheless, the relative contents of exogenously expressed pumps can be roughly estimated by the Fluorescence Size-Exclusion Chromatography (FSEC) due to the GFP at N-term as we did in Nakanishi et al., 2020, *Cell Rep*. Compared to NKA (1.5 a.u.) and gHKA (1.6 a.u.), ngHKA (4.6 a.u.) and its single mutants (K794A: 26.6 a.u., K794S: 14.7 a.u.), have higher maximum ATPase activity (V_{max}) due to their higher expression level. The SPWC mutant has similar expression than NKA (1.8 a.u.). This is briefly mentioned in Method section.

"The membrane fractions (1 mg/ml) used for the ATPase measurement were solubilized with 1% LMNG in the presence of 1 mM BeSO₄, 3 mM NaF, 1 mM MgCl₂, 1% glycerol

and 40 mM MES/Tris, pH 6.5 for 20 min on ice. After removing insoluble materials by ultracentrifugation, the supernatant was analyzed by fluorescence size-exclusion chromatography (FSEC) monitoring GFP fluorescence (Ex 495 nm, Em 520 nm). The peak fluorescence values (in arbitrary unit, a.u.) serve as a measure of the relative amount of enzyme in the membrane fraction. Compared to NKA (peak value 1.5 a.u.) and gHKA (1.6 a.u.), ngHKA WT (4.6 a.u.) and its single mutants, K794A (26.6 a.u.) and K794S (14.7 a.u.) show higher expression level, consistent with their relatively high specific activities in the membrane fraction. The SPWC mutant, in contrast, had similar expression than NKA (1.8).”

5) Figure 1 is very pixelated especially the alignment, I assume this is a conversion to pdf issue.

Yes. This seems to be a conversion issue for initial review. It will be corrected before publication.

6) When looking at the validation reports all structures have worse clashscores than expected, this is most apparent for 7X20, is there a reason for this value being worse in this structure?

We performed crystal refinement based on R_w/R_f value and Ramachandran plot and have not paid much attention for Clashscores. Although the electron density map is a bit noisy in some parts, especially in loop structures, we could build reasonable amino acid model based on the electron density (we checked all residues and their connections by eye). The figure shows amino acid residues that are found to show Clash overlap of more than 0.8 Å in the validation report (in 7x20, non-gastric alpha2 WT, E2P state). Given that the amino acids with bad clashscores (red spheres) are scattered in the molecule’s “surface”, it is likely that the poor EM density is responsible for such high clash value. Therefore, we think that the cation-binding sites important to this paper’s conclusion is well resolved and appropriately modeled.

7) The complementary mutations to change the nature of the pump are interesting. Are there

any examples in this family where nature has already made similar changes to repeat this?

We agree it would be interesting. Obviously, we are working on figuring out NKAs with different stoichiometry and their importance for physiology. We are not aware of HKAs with different stoichiometry than 1:1.

Reviewer #3 (Remarks to the Author)

Minor comments:

I.42: "... a process **promoting** chronic respiratory ..."

Corrected.

I.58: Neither here nor in the methods it is explained why a rat NKA β subunit was used and instead of a ngHKA β subunit.

There is not such a thing as the ngHKA β subunit. The most probable natural partner is the NKA β . In any case, we tested expression of the ngHKA α subunit with the rat NKA β 1-, NKA β 3- and gastric β -subunit for co-expression with the rat non-gastric HKA α -subunit, and found that coexpression with the rat NKA β 1-subunit gave the highest expression level as well as K⁺-dependent ATPase activity in the membrane fraction. We don't think it is necessary to mention this in the article, as we did not repeat each expression with the other β too many times.

I.203: When 25 mV lead to a two-fold reduction, why does a $(-85 \text{ mV}) - (-51 \text{ mV}) = (-34 \text{ mV})$ shift lead to a ~4-fold reduction? (Why not ~2.7-fold?)

The reviewer is right. We apologize for the oversight. It has been corrected.

I.249: "Compared to E2, this is accompanied..." What means "this"? The (overall) process?

Thanks for the helpful comment. Here "this" means coordination of site II Na by Glu346 and TM4 residues. We revise this sentence as follows.

"When comparing this conformation, to the E2 state structure, it is clear that Na⁺ coordination occurred at site II needs to be accompanied by a large conformational change of the SPWC structure (**Fig. 8, Supplementary Movie 1**), resembling the changes occurring in NKA"

I.264-268: When site III is occupied first (I.245), the displacement of 5.6 Å and P-domain movement will occur before sites I and II are occupied. But what prevents then an untimely autophosphorylation?

We think that like in E2P state, the A domain blocks autophosphorylation until it's movement is induced by displacement of TM1-TM4, which is triggered by binding to site II, the final ion-binding step. In other words, site III Na⁺ binding alone is insufficient for the P domain inclination. Conformational change of the A domain, which is induced by the TM1-2 and TM4, which is induced by the site I and II Na⁺, which is cooperatively supported by the site III Na⁺,

are required for the autophosphorylation.

I.273: "... that **the side chain of** 943 faces ..."

Corrected.

I.303: "... Ile943, **prevents** Na⁺ binding ..."

Corrected.

I.356: "... extrapolation **of** our findings ..."

Corrected.

I.582: "... **Å²** ..."

Corrected.

I.633: "The **amount of charge transferred was calculated** by integrating ..."

Corrected.

I.752: "... (NKA, **black**) ..."

Corrected.

I.764: "... **NKA** crystal structure (wheat ..."

Corrected.

I.771: "... ngHKA WT (**blue**) ..."

Corrected.

I.772: "... maximum **measured** K⁺-activated ..."

Corrected.

I.781: Here and elsewhere (Figs. 2b, 3a, 3e, 4a, 4c) the length and position of the bars indicating application of substrates seem to be somewhat erratic in several places. Please replace by bars of correct length and position.

Apologies, corrected.

I.799: "... Lys794Ala (n = 7) **and WT** could not ..."

Corrected.

I.841: "... ouabain-sensitive (**OS**) current ..."

Corrected.

I.852: "... ATPase **activity** in ..."

Corrected.

I.854: "... (orange) mM **Na⁺**."

Thanks, corrected.

I.863: In case of $3\text{Na}^+\text{-E1-ATP}$ (panel h) the β subunit is not shown in grey.

Certainly, thanks for the remarks. We revise this figure.

I.866: "... the **3** bound Na^+ ..."

Corrected.

I.876: "... (e) **bent** (purple, ..."

Corrected.

I.881: "Dashed" instead of "dotted" lines?

Corrected.

I.920 "... activites of **various ATPases studied**"

Corrected.